# Antimicrobials from a feline commensal bacterium inhibit skin infection by drug-resistant *S. pseudintermedius*

Alan M O'Neill[1], Kate A Worthing[2], Nikhil Kulkarni[1], Fengwu Li[1], Teruaki Nakatsuji[1], Dominic McGrosso[3,4], Robert H Mills[3,4], Gayathri Kalla[5], Joyce Y Cheng[1], Jacqueline M Norris[6], Kit Pogliano[5], Joe Pogliano[5], David J Gonzalez[3,4], Richard L Gallo[1]*

[1]Department of Dermatology, University of California, San Diego, San Diego, United States; [2]College of Veterinary Medicine, University of Arizona, Oro Valley, United States; [3]Department of Pharmacology, University of California, San Diego, San Diego, United States; [4]Skaggs School of Pharmacy and Pharmaceutical Sciences, University of California, San Diego, San Diego, United States; [5]Division of Biological Sciences, University of California, San Diego, San Diego, United States; [6]Sydney School of Veterinary Science, University of Sydney, Sydney, Australia

**Abstract** Methicillin-resistant *Staphylococcus pseudintermedius* (MRSP) is an important emerging zoonotic pathogen that causes severe skin infections. To combat infections from drug-resistant bacteria, the transplantation of commensal antimicrobial bacteria as a therapeutic has shown clinical promise. We screened a collection of diverse staphylococcus species from domestic dogs and cats for antimicrobial activity against MRSP. A unique strain (*S. felis* C4) was isolated from feline skin that inhibited MRSP and multiple gram-positive pathogens. Whole genome sequencing and mass spectrometry revealed several secreted antimicrobials including a thiopeptide bacteriocin micrococcin P1 and phenol-soluble modulin beta (PSMβ) peptides that exhibited antimicrobial and anti-inflammatory activity. Fluorescence and electron microscopy revealed that *S. felis* antimicrobials inhibited translation and disrupted bacterial but not eukaryotic cell membranes. Competition experiments in mice showed that *S. felis* significantly reduced MRSP skin colonization and an antimicrobial extract from *S. felis* significantly reduced necrotic skin injury from MRSP infection. These findings indicate a feline commensal bacterium that could be utilized in bacteriotherapy against difficult-to-treat animal and human skin infections.

*For correspondence:
rgallo@health.ucsd.edu

## Introduction

Skin is colonized by hundreds of diverse bacterial species that exist within a complex and dynamic chemical landscape. These chemical interactions can play important roles in skin health, immune education and protection against pathogen colonization and infection (*Sanford and Gallo, 2013*). The composition of the skin microbial community of humans and animals varies extensively, in part due to different skin habitats, that is increased hair density in animals, as well as more subtle differences in the chemical and biological conditions of the skin (*Grice and Segre, 2011*; *Ross et al., 2018*). Overall, the human microbial skin community is distinct from and significantly less diverse than that of both wild and domestic animals (*Ross et al., 2018*). Human skin is generally dominated by few taxa present at high abundance for example cutibacteria, streptococci, and staphylococci, whereas canine skin harbors a more equally distributed and diverse group of taxa (*Song et al., 2013*). Naturally, close contact between humans and animals can be a source for microbial transmission (*Frana et al., 2013*;

*Lai et al., 2017*). Although it remains to be determined if shared taxa are stable over time, there are reports that exposure to pets early in life can be protective against atopic disease in later life (*Mandhane et al., 2009*).

In contrast, there are also many documented cases of human staphylococcal infection from epidemiological exposure to dogs (*Somayaji et al., 2016*). Companion animals can act as reservoirs for methicillin-resistant *S. aureus* (MRSA) and more commonly, *S. pseudintermedius* (MRSP), with both species sharing many common invasion and virulence factors (*Garbacz et al., 2013*). The zoonotic significance of *S. pseudintermedius* may have been previously underestimated because it was frequently misidentified as *S. aureus* in human wound infections (*Börjesson et al., 2015*). More advanced diagnostic techniques such as matrix-assisted laser desorption/ionization-time of flight (MALDI-TOF) mass spectrometry have led to increased detection of human *S. pseudintermedius* infections (*Ference et al., 2019*). Colonization of *S. pseudintermedius* is a contributing factor in canine atopic dermatitis (AD). Interestingly, the prevalence of AD in humans and AD in dogs are similar (10–15% in US) and present with remarkably similar immunological and clinical manifestations (*Marsella and Girolomoni, 2009*; *Silverberg, 2019*). Likewise, several studies have reported a decrease in the microbiome diversity of AD and increased colonization of *S aureus* in humans and *S. pseudintermedius* in dogs (*Fazakerley et al., 2009*; *Nakatsuji and Gallo, 2019*; *Older et al., 2020*). In human AD, *S. aureus* was identified in higher relative abundances during disease flares (*Kong et al., 2012*). Similarly, the relative abundance of *S. pseudintermedius* was also shown to increase with disease flares in canine AD (*Bradley et al., 2016*). Common treatment modalities exist for both diseases. Dilute bleach baths are a common antiseptic treatment for AD, with the goal of reducing the carriage of staphylococci (*Banovic et al., 2018*; *Chopra et al., 2017*). However, its effectiveness as an antibacterial agent is controversial (*Sawada et al., 2019*).

An alternative and promising approach is not to disrupt but to re-establish the community of microbes on the skin that promote health. To do this our group and others have identified naturally occurring commensal species on healthy human skin that express antimicrobial activity against pathogens. A recent successful example of this approach is the discovery and use of commensal staphylococcus strains that produce lantibiotics that when applied to skin of patients with AD reduced *S. aureus* counts and improved clinical outcome (*Nakatsuji et al., 2021a*; *Nakatsuji et al., 2021b*). Other studies have also identified antimicrobial activity in some staphylococcus strains belonging to species of *S. lugdunensis* (*Zipperer et al., 2016*), *S. epidermidis* (*Cogen et al., 2010*), and *S. capitis* (*O'Neill et al., 2020*). In contrast, very little is known regarding the antimicrobial activity of staphylococci derived from the animal commensal microbiome and their clinical potential against skin infection. Here, we identified the strain *S. felis* C4, a potent antimicrobial isolate from feline skin that inhibited the growth of MRSP in vitro and in vivo. *S. felis* C4 produced a thiopeptide bacteriocin and several α-helical amphipathic peptides with antimicrobial and anti-inflammatory activity. This discovery represents a potential new bacteriotherapeutic for human and animal skin diseases associated with *S. pseudintermedius* colonization and infection.

## Results

### A screen of animal-derived staphylococcus isolates identifies a feline skin commensal bacterium with broad-spectrum antimicrobial activity

We sought to determine whether commensal staphylococci collected from the skin, nasal, oral and perineal sites of companion dogs and cats exhibit antimicrobial activity against methicillin-resistant *S. pseudintermedius* (MRSP) ST71 (*Figure 1A*Ma et al., 2020; *Worthing et al., 2018b*). Fifty-eight staphylococcus isolates across the coagulase-positive (CoPS) and coagulase-negative (CoNS) groups were screened, including validated antimicrobial strains of human origin, *S. hominis* A9 (*Nakatsuji et al., 2017*) and *S. capitis* E12 (*O'Neill et al., 2020*) and a non-active *S. aureus* 113 negative control strain (*Figure 1B*). The animal test isolates were screened for antimicrobial activity by live co-culture on agar plates or in the presence of sterile conditioned supernatant, as illustrated in *Figure 1A*. Amongst all test isolates, five strains demonstrated greater than 80 % inhibition of *S. pseudintermedius* growth (dashed line) across all three different dilutions of supernatant (1:1, 1:4, 1:8) (*Figure 1C*). Surprisingly, these strains exhibited greater potency compared to the positive control *S. hominis* A9 supernatant (indicated by black circle), which inhibited growth of *S. pseudintermedius* in a 1:1 dilution,

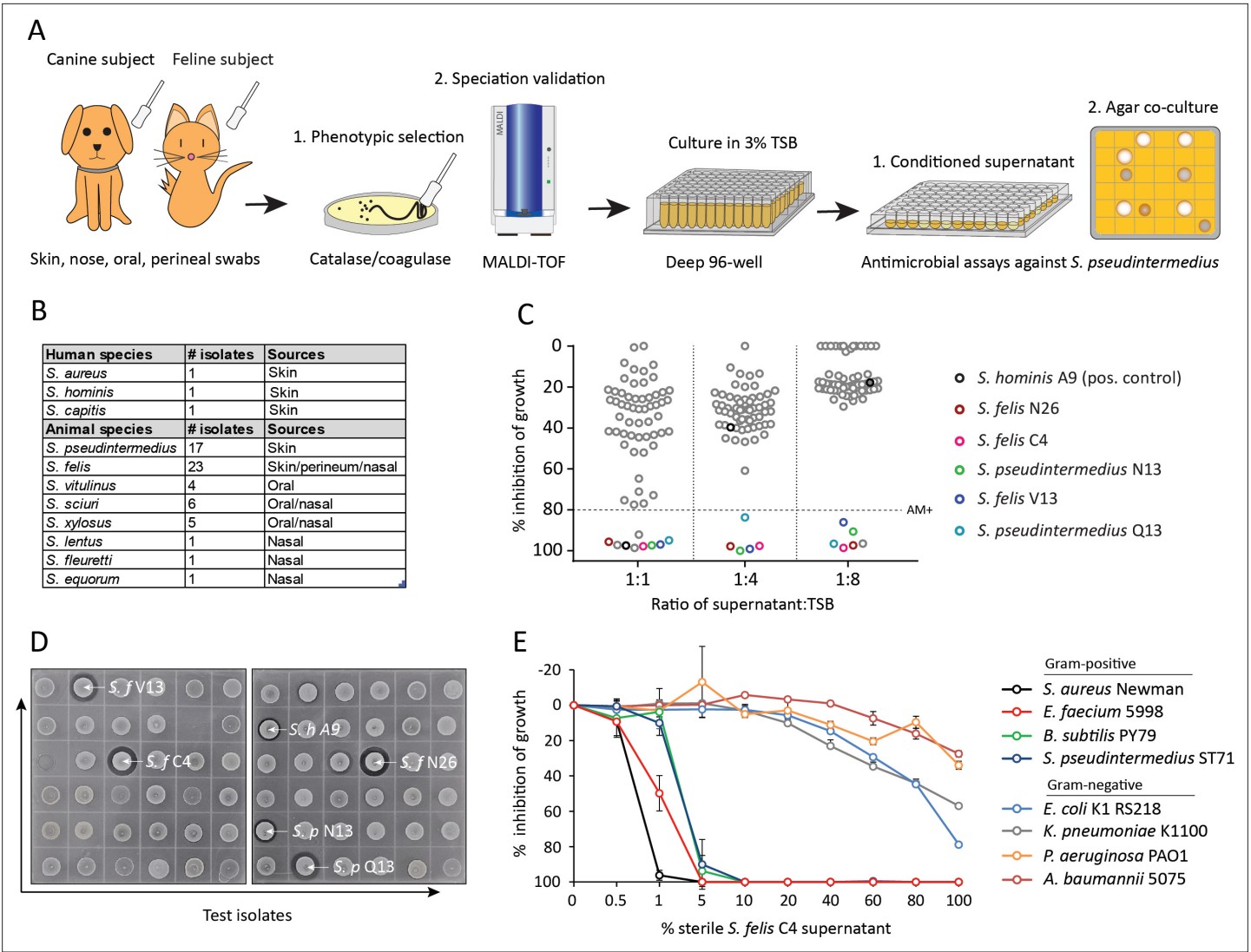

**Figure 1.** Screening and discovery of a feline skin commensal bacterium that inhibits drug-resistant gram-positive pathogens. (**A**) Illustration of the selection and screening strategy of animal-derived staphylococci against the growth of methicillin-resistant *S. pseudintermedius* (MRSP) ST71 in liquid culture and agar co-culture assays. (**B**) The panel of 58 feline and canine isolates selected for antimicrobial testing, as well as human-derived *S. hominis* A9 and *S. capitis* E12 positive control antimicrobial strains and the non-antimicrobial *S. aureus* 113 negative control. (**C**) Inhibition of *S. pseudintermedius* ST71 growth by OD600, relative to TSB control at 100%, after 18 h incubation in 50%, 25%, or 12.5 % (1:1, 1:4, 1:8 ratio) sterile conditioned supernatant from all staphylococci isolates. Greater than 80 % inhibition of growth was considered antimicrobial (AM+). (**D**) Images of the agar co-culture assay showing zone of inhibition (black circle surrounding colony) produced by all staphylococci test isolates against *S. pseudintermedius* ST71, including *S. felis* C4, N26, V13 (*S. f* C4, *S. f* N26, *S. f* V13), *S. pseudintermedius* N13 and Q13 (*S. p* N13 and *S. p* Q13), and positive control *S. hominis* A9 (*S. h* A9, all indicated by arrows). (**E**) Inhibition of bacterial growth by OD600, normalized to TSB alone at 100%, against select gram-positive and gram-negative pathogens after 18 h incubation (48 h incubation for *E. faecium*) in the presence of increasing amounts of sterile conditioned supernatant from *S. felis* C4 overnight growth. Error bars indicate SEM. Representative of three separate experiments.

The online version of this article includes the following figure supplement(s) for figure 1:

**Source data 1.** Source data for *Figure 1C* (% growth of *S.p* ST71 in SN of animal isolates) and *Figure 1E* (% growth of gram positive and gram negative pathogens in *S. felis* C4 SN).

**Source data 2.** Source data for *Figure 1D* (labeled and unlabeled images of the antimicrobial agar assay).

**Figure supplement 1.** Generation of a partially purified antimicrobial extract from *S. felis* C4.

**Figure supplement 1—source data 1.** Source data for *Figure 1—figure supplement 1A* (labeled and unlabeled images of the antimicrobial agar assay).

**Figure supplement 1—source data 2.** Source data for *Figure 1—figure supplement 1B* (labeled and unlabeled images of the antimicrobial agar assay).

*Figure 1 continued on next page*

*Figure 1 continued*

**Figure supplement 1—source data 3.** Source data for *Figure 1—figure supplement 1C* (labeled and unlabeled images of the antimicrobial agar assay).

**Figure supplement 1—source data 4.** Source data for *Figure 1—figure supplement 1D* (labeled and unlabeled images of the silver-stained bacterial protein before and after extraction).

**Figure supplement 2.** *S. felis* C4 supernatant and extract disrupt *S. pseudintermedius* biofilm.

**Figure supplement 2—source data 1.** Source data for *Figure 1—figure supplement 2A*,B (raw values for the biofilm inhibition).

but not at 1:4 or lower. Amongst the five positive hits, three were identified as *S. felis* and two *S. pseudintermedius*. In the second independent antimicrobial assay, all five isolates including positive control *S. hominis* A9, produced an observable zone of inhibition against *S. pseudintermedius* during live co-culture on agar (*Figure 1D*). The two feline *S. felis* species (C4, N26 labelled with white arrows) produced the largest inhibitory zones, extending 3.0–3.3 mm outward from the edge of the growing colony. The *S. felis* C4 strain was chosen for further analysis, as it demonstrated potent activity and was isolated from healthy skin. To investigate the significance and selectivity of the *S. felis* antimicrobial supernatant, we tested its capacity to inhibit the growth of other clinically relevant, gram-positive and gram-negative pathogens (of which several belong to the clinically-relevant ESKAPE group). Of the four gram-negative strains tested, only moderate inhibition was demonstrated after 18 hr incubation with 80–100% of the *S. felis* C4 supernatant (*Figure 1E*). In contrast, bacterial culture in the presence of just 1–5% of *S. felis* C4 supernatant was sufficient to inhibit >80% growth of all four gram-positive organisms, including *S. pseudintermedius*, *E. faecium*, *B. subtilis,* and *S. aureus*.

Of the three antimicrobial *S. felis* isolates, only the C4 supernatant retained activity after precipitation with 75 % ammonium sulfate (AS) (*Figure 1—figure supplement 1A*). Moreover, 75 % AS was highly effective in precipitating the antimicrobial factor(s) from the C4 supernatant since no activity could be visualized in the non-precipitate fraction (*Figure 1—figure supplement 1B*). This effect was also achieved with a simpler extraction by n-butanol. The antimicrobial butanol extract remained active up to 1 week at room temperature (RT) and was stable after boiling (*Figure 1—figure supplement 1C*). As expected, the butanol extraction provided a partially purified and enriched antimicrobial fraction compared to that of the crude supernatant and AS precipitation (*Figure 1—figure supplement 1D*). Therefore, this *S. felis* C4 extract obtained via n-butanol extraction was then adopted for further experiments. The *S. felis* extract was found to be effective against multiple clinical isolates of *S. aureus* and *S. pseudintermedius* with a MIC range of 0.4–6.25 µg/ml (*Figure 1—figure supplement 1E*). In contrast, some CoNS strains, including *S. hominis* and *S. lugdunensis* were less sensitive and had a higher MIC of 12.5 µg/ml.

We next evaluated the activity of *S. felis* C4 extract to disrupt bacterial biofilms. Biofilm formation is considered an important determinant of staphylococci virulence and is associated with increased skin colonization and severity of disease (*Di Domenico et al., 2018*; *Kwiecinski et al., 2015*). Reports have shown that most clinically-derived *S. pseudintermedius* strains are biofilm producers (*Singh et al., 2013*). A 4 hr preformed biofilm of *pseudintermedius* ST71 showed a significant decrease in crystal violet (CV) staining over time, when incubated with 100 % conditioned supernatant of *S. felis* C4, indicating biofilm disruption and degradation (*Figure 1—figure supplement 2A*). The *S. felis* C4 extract had anti-biofilm activity similar to crude conditioned supernatant, with biofilm mass reduced by 48 % at 250 µg/ml and 58 % at 500 µg/ml (*Figure 1—figure supplement 2B*).

## Purification and identification of PSMβ peptides as antimicrobial products of *S. felis* C4

To determine the nature of the antimicrobial product produced by *S. felis* C4, sterile culture supernatant was purified by HPLC. This yielded two major peaks that eluted at 44% and 47% acetonitrile (*Figure 2A*). Anti-*S. pseudintermedius* activity was predominantly associated with fraction 32 which eluted at 47 % acetonitrile (*Figure 2B*). SDS PAGE and protein silver staining of the active and inactive fractions revealed a unique band of roughly 5 kDa in size in the active fraction 32 (*Figure 2C*). To determine if this small protein was responsible for antimicrobial activity, gel slices of the fraction 32 lane corresponding to small, medium and larger proteins ( ≤ 5 kDa, 5–20 kDa and 20–50 kDa, respectively) were excised and extracted by acetone precipitation as previously described (*Botelho et al.,*

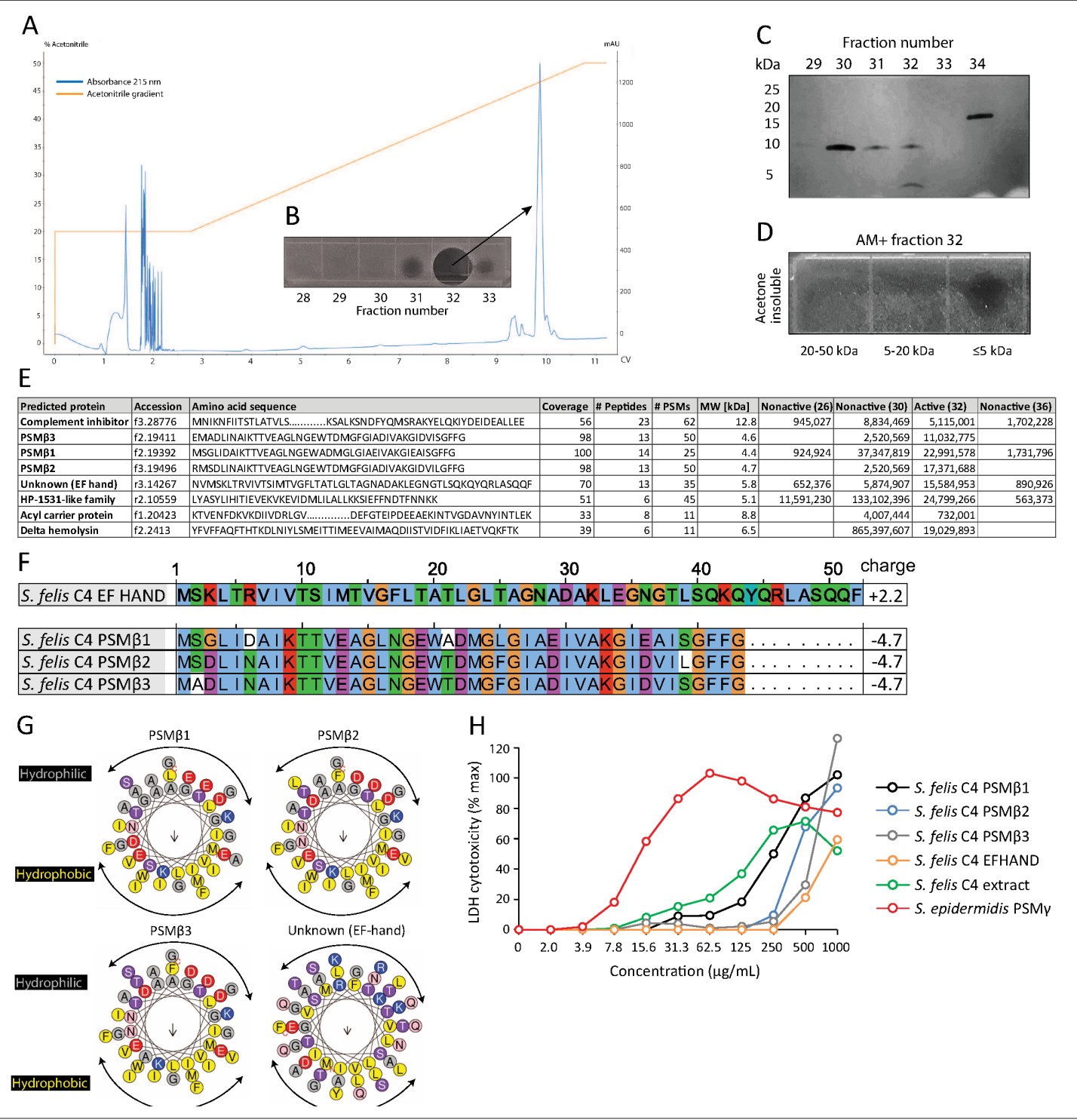

**Figure 2.** HPLC purification yields an antimicrobial fraction from *S. felis* C4 supernatant. (**A**) Reverse-phase high-performance liquid chromatography (HPLC) elution profile from sterile supernatant of *S. felis* C4 strain loaded onto a C8 column. (**B**) Inset image of antimicrobial activity exhibited by fraction 32 against *S. pseudintermedius* ST71 corresponding to the indicated peak. (**C**) Silver stain of total protein content in the different fractions indicated. (**D**) Radial diffusion assay of antimicrobial activity of the AM + fraction 32 after extraction and acetone precipitation of proteins within different sized silver stain gel fragments. (**E**) Mass spectrometry (MS) table of the top eight peptide hits obtained from HPLC fractions that were active (fraction 32) or inactive (26, 30, 36) against *S. pseudintermedius* ST71. (**F**) ClustalW multiple amino acid sequence alignment of all three *S. felis* C4 genetically-encoded PSMβ peptides with predicted net charge at pH 7.4 (Prot pi) and amino acid sequence of a EF-hand domain-containing peptide with unknown function. (**G**) Alpha helical wheel plots of *S. felis* C4 PSMβ1–3 and EF-HAND domain peptide, indicating conserved α-helical, amphipathic-like structures with

*Figure 2 continued on next page*

*Figure 2 continued*

indicated hydrophobic yellow residues confined to one side (indicated by arrow) and gray hydrophilic residues on the opposing side. (**H**) LDH release in NHEKs after 24 h treatment with increasing concentrations of *S. felis* C4 extract, *S. felis* C4 EF-HAND synthetic peptide, *S. felis* formylated synthetic PSMβ1, PSMβ2, PSMβ3, or positive control cytotoxic PSMγ from *S. epidermidis*. Percentage (%) cytotoxicity measured by maximum LDH release into supernatant collected after untreated cell freeze thaw.

The online version of this article includes the following figure supplement(s) for figure 2:

**Source data 1.** Source data for *Figure 2B* (labeled and unlabeled images of the antimicrobial agar assay from HPLC fractions).

**Source data 2.** Source data for *Figure 2C* (labeled and unlabeled images of the silver-stained bacterial protein after HPLC purification).

**Source data 3.** Source data for *Figure 2D* (labeled and unlabeled images of the antimicrobial agar assay from active fraction 32 and nonactive fraction 28 after acetone precipitation).

**Source data 4.** Source data for *Figure 2E* (raw and annotated MS data) and *Figure 2H* (LDH cytotoxicity values).

*2010*; *Zhang et al., 2015*). Only the ≤5 kDa band demonstrated antimicrobial activity after incubation with *S. pseudintermedius* (*Figure 2D*), thereby suggesting the likely candidate to be a small peptide. Mass spectrometry (MS) analysis of the top eight hits in the active and non-active HPLC fractions identified several putative small antimicrobial peptides (AMP), representing the phenol soluble modulin beta (PSMβ1–3) and gamma (PSMγ, aka delta-hemolysin) families and a peptide of unknown function containing the EF-HAND domain, common amongst some antimicrobial $Ca^{2+}$ binding proteins, such as S100A8/S100A9 (*Chazin, 2011*; *Figure 2E*). Whole genome analysis of the *S. felis* C4 strain confirmed the presence of three PSMβ-encoding genes (*Figure 2F*). Based on sequence similarities to mammalian cationic AMPs such as cathelicidin LL-37 the PSMβ and EF-HAND domain peptides were predicted to have α-helical amphipathic structure (*Figure 2G*). Synthetic N-formylated versions of all three *S. felis* C4 PSMβ1–3 and the EF-hand domain containing peptide were then generated and added to primary cell cultures of normal human keratinocytes (NHEK) for 24 hr to determine the level of cytotoxicity as measured by lactate dehydrogenase (LDH) release. The positive control cytotoxic PSMγ induced maximal cell death with 100 % LDH release at 62.5 ug/ml, yet at this concentration none of the PSMβ peptides nor the crude *S. felis* extract exceeded 20 % LDH release (*Figure 2H*), suggesting that *S. felis* C4 PSMβs are not cytotoxic.

## *S. felis* C4 extract and PSMβ peptides exhibit anti-inflammatory activity by suppressing TLR-mediated inflammation

Unlike the well-characterized cytolytic and inflammatory activities of PSMα, a defined role for PSMβ in mediating host interactions has been largely unexplored (*Da et al., 2017*). To investigate the potential effects of *S. felis* C4 on the host immune response, we stimulated cells with various TLR agonists in the presence or absence of the extract or the individual PSMs and measured inflammatory gene expression. NHEKs were treated with *S. felis* PSMβ2, PSMβ3, extract, DMSO control alone, or each in combination with the TLR2 agonist MALP-2 (200 ng/µl) or the TLR3 agonist Poly I:C (0.4 µg/ml) for 4 hr. Neither PSMβ nor the extract resulted in any detectable increase in gene expression, whereas MALP-2 significantly increased the expression of hBD-2, and Poly I:C significantly increased CXCL10 and IL-6 expression (*Figure 3A–C*). Interestingly, these TLR-mediated responses were significantly reduced during co-treatment with *S. felis* C4 PSMβ or extract. This result was confirmed by ELISA, showing that PSMβ2 had a significant effect on suppressing CXCL10 secretion in NHEKs after 24 hr co-treatment (*Figure 3D*). To determine if this interaction is specific for epithelial cells, we also stimulated human THP-1 macrophage-like cells with MALP-2, or the TLR4 agonist LPS and found that IL-6 and TNFα expression was decreased during co-treatment with *S. felis* C4 PSMβ2 (*Figure 3—figure supplement 1*). The addition of *S. felis* C4 PSMβ2 to NHEK activated by poly I:C demonstrated that PSMβ2 inhibited phosphorylation of TBK1 and IRF3 at 15 min post-stimulation with the peptide (*Figure 3E*). This inhibition of inflammatory target gene and kinase activity was further evaluated by the analysis of changes in global gene expression using RNA-Seq analysis of NHEKs stimulated with Poly I:C, with and without PSMβ2 at 4 hr and 24 h. Gene ontology (GO) analyses revealed the significant down-regulation of several gene clusters associated with 'immune effector process' and 'type I IFN signaling' at 4 h during co-treatment of PSMβ2 and Poly I:C (*Figure 3F*). A heatmap of selected genes within the 'Immune response' GO term at 4 h post-treatment further highlighted the suppressive effect, and also importantly showed that PSMβ2 treatment alone did not induce an immunological

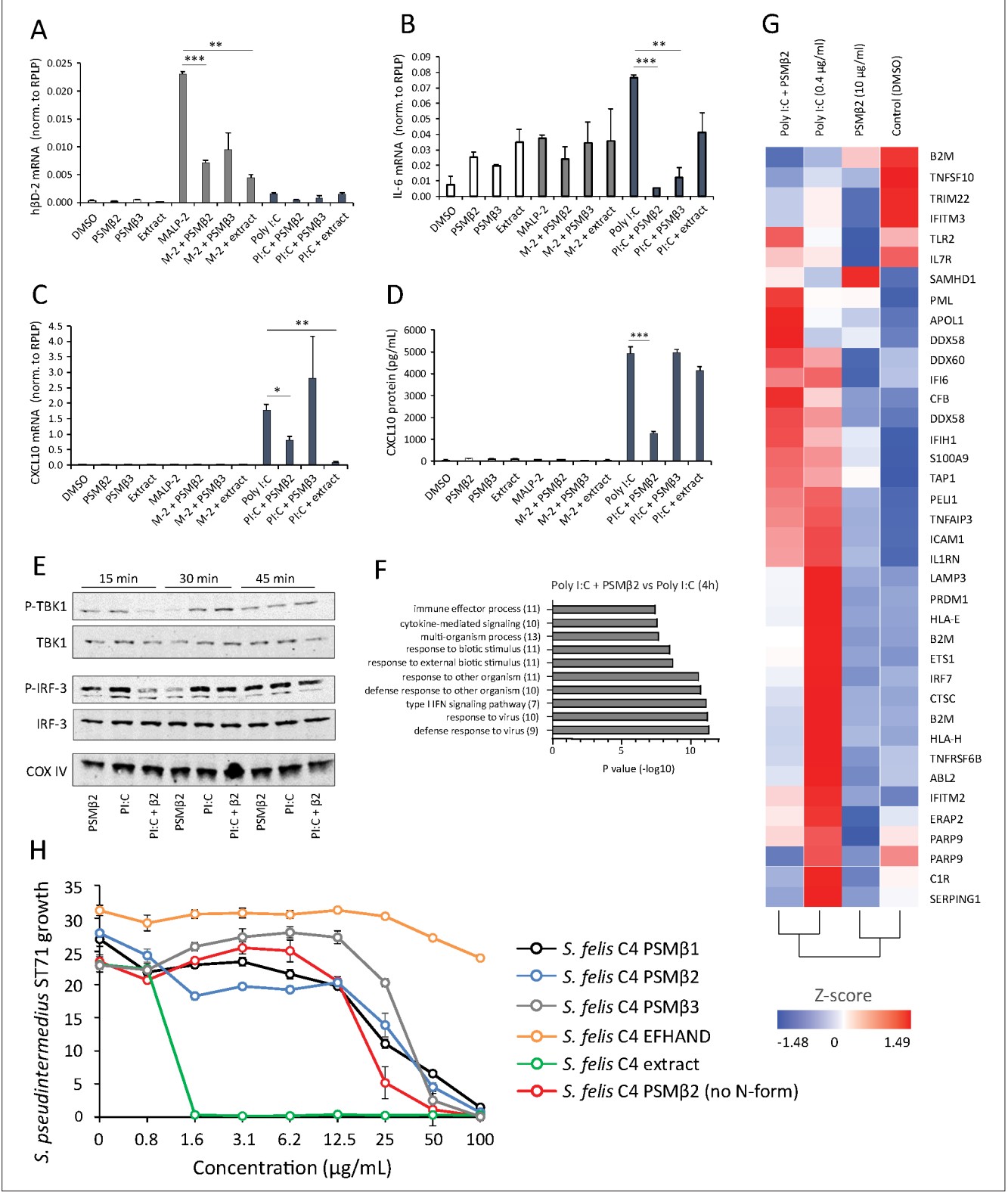

**Figure 3.** Antimicrobial *S. felis* C4 extract and PSMβ suppress TLR-mediated inflammation. mRNA transcript abundance of hBD2 (**A**), IL-6 (**B**) and CXCL10 (**C**) as measured by qPCR in NHEKs stimulated with or without TLR2/6 agonist MALP-2 (M-2) (200 ng/ml) or TLR3 agonist Poly I:C (PI:C) (0.4 µg/ml) in the presence or absence of *S. felis* C4 extract, PSMβ2 or PSMβ3 (10 µg/ml) or DMSO control (0.1%) at 4 hr post-treatment. (**D**) Quantification of CXCL10 protein by ELISA from the supernatant of NHEKs stimulated with MALP-2 or Poly I:C in the presence or absence of *S. felis* extract, PSMβ2, PSMβ3 or DMSO control 24 hr post-treatment. (**A–D**) Error bars indicate SEM. One-way ANOVA with multiple corrections (Tukey's correction) was

*Figure 3 continued on next page*

*Figure 3 continued*

performed. p values: * p < 0.05; ** p < 0.01; *** p < 0.001. (**E**) Time-course of the TLR3 signaling cascade by immunoblot of phosphorylated TBK1 (P-TBK1) and IRF-3 (P-IRF-3) proteins after stimulation of NHEKs with Poly I:C (PI:C), PSMβ2 (β2), or co-treatment with Poly I:C and PSMβ2 (PI:C+β2). (**F**) Gene ontology (GO) pathway analysis of genes downregulated in NHEKs after 4 hr co-treatment with Poly I:C and PSMβ2 versus treatment with Poly I:C alone. (**G**) Hierarchical clustering and Heatmap visualization of selected genes from GO enriched 'immune response' pathway (1.5-fold change) 4 hr post-treatment with DMSO, PSMβ2 or Poly I:C alone or with Poly I:C and PSMβ2 cotreatment. (**H**) Growth of *S. pseudintermedius* ST71 (OD600 nm) after 18 hr incubation with increasing concentrations of *S. felis* C4 extract, formylated peptides PSMβ1, PSMβ2, PSMβ3, EFHAND domain-containing peptide or non-formylated PSMβ2. Error bars indicate SEM. Representative of two independent experiments.

The online version of this article includes the following figure supplement(s) for figure 3:

**Source data 1.** Source data for *Figure 3A–C* (gene expression values measured by qPCR), *Figure 3D* (secreted CXCL10 values measured by ELISA), *Figure 3F* (gene list for GO terms) and *Figure 3H* (growth of *S.p* ST71 in the presence of peptides and extract).

**Source data 2.** Source data for *Figure 3E* (labeled and unlabeled western blots of (P) TBK1, (P) IRF3 and COXIV).

**Figure supplement 1.** *S. felis* C4 PSMβ2 reduces TLR2- and TLR4-stimulated transcripts in THP-1 macrophages.

**Figure supplement 2.** *S. felis* C4 PSMβ2 downregulates transcripts associated with cytokine signaling.

**Figure supplement 3.** PSMβ do not exhibit synergistic antimicrobial activity.

response in NHEKs (*Figure 3G*). However, when NHEKs were treated with PSMβ2 in the absence of an inflammatory stimulus, we identified the downregulation of genes within GO terms such as "cytokine-mediated signaling" and "pathogenic *E. coli* infection" (*Figure 3—figure supplement 2*), suggesting that exposure to PSMβ2 primes cells to dampen potential inflammatory mediators in response to TLR ligands. This contrasted with PSMβ2-upregulated genes which were mostly associated with biosynthetic pathways including lipid and amino acid metabolism (data not shown).

The antimicrobial activity of all three synthetic *S. felis* C4 PSMβ (PSMβ1–3) peptides in inhibiting *S. pseudintermedius* was observed to occur at a concentration of 50 µg/ml and the synthetic EF-HAND domain-containing peptide did not inhibit bacterial growth at concentrations up to 200 µg/ml (*Figure 3H*). In contrast, the native extract prepared from *S. felis* supernatant was found to be more potent than the synthetic PSM when added individually (*Figure 3H*) or in combination (*Figure 3—figure supplement 3*). A non-N-formylated version of PSMB2 exhibited similar activity to the N-formylated PSMβ2 version (*Figure 3H*). This disparity in the potency of the synthetic peptides compared to the native extract suggested that another antimicrobial molecule is produced by *S. felis* C4 that was not detected in the prior analyses.

To investigate this, we conducted further genome analysis using the web server anti-SMASH to identify the potential presence of secondary metabolites encoded from biosynthetic gene clusters (BCG) (*Weber et al., 2015*). Interrogation of the *S. felis* C4 genome revealed a BCG with similarity to microccocin P1 from *Macrococcus caseolyticus* (*Figure 4A*). Microccocin P1 is a macrocyclic antibiotic that inhibits ribosome translation in gram-positive bacteria (*Figure 4B Carnio et al., 2000*; *Ciufolini and Lefranc, 2010*). To determine if microccocin P1 could be present within the *S. felis* C4 extract, we compared the C8 HPLC elution profile of synthetic microccocin P1 to the native *S. felis* C4 extract and identified that the synthetic peptide eluted as a single major peak at 59 % acetonitrile in fractions 27 and 28 (*Figure 4C*). These fractions were active against *S. pseudintermedius* ST71 by agar radial diffusion (*Figure 4C* inset). Purification by HPLC of the *S. felis* C4 extract revealed a similar major peak that eluted at 59 % acetonitrile in fractions 27 and 28 (*Figure 4D*) and this also was active against *S. pseudintermedius* ST71 (*Figure 4D* inset). Mass spectrometry analysis of this elution fraction from *S. felis* C4 extract showed a peptide with mass of 1,144 daltons which was identical to synthetic microccocin P1 (*Figure 4E* and *Figure 4—figure supplement 1*). Furthermore, both synthetic microccocin P1 and the *S. felis* C4 antimicrobial activity could be extracted with butanol (*Figure 4F*). These observations strongly suggested that *S. felis* produces a peptide similar to microccocin P1 and this peptide together with the previously identified PSMs contributed to the highly potent antimicrobial activity of *S. felis* C4.

## Antimicrobial *S. felis* C4 inhibits bacterial translation and disrupts the membrane

Next, we sought to better understand how the antimicrobial action of *S. felis* C4 negatively affects bacterial physiology. Given the selective nature of the *S. felis* C4 supernatant against gram-positives

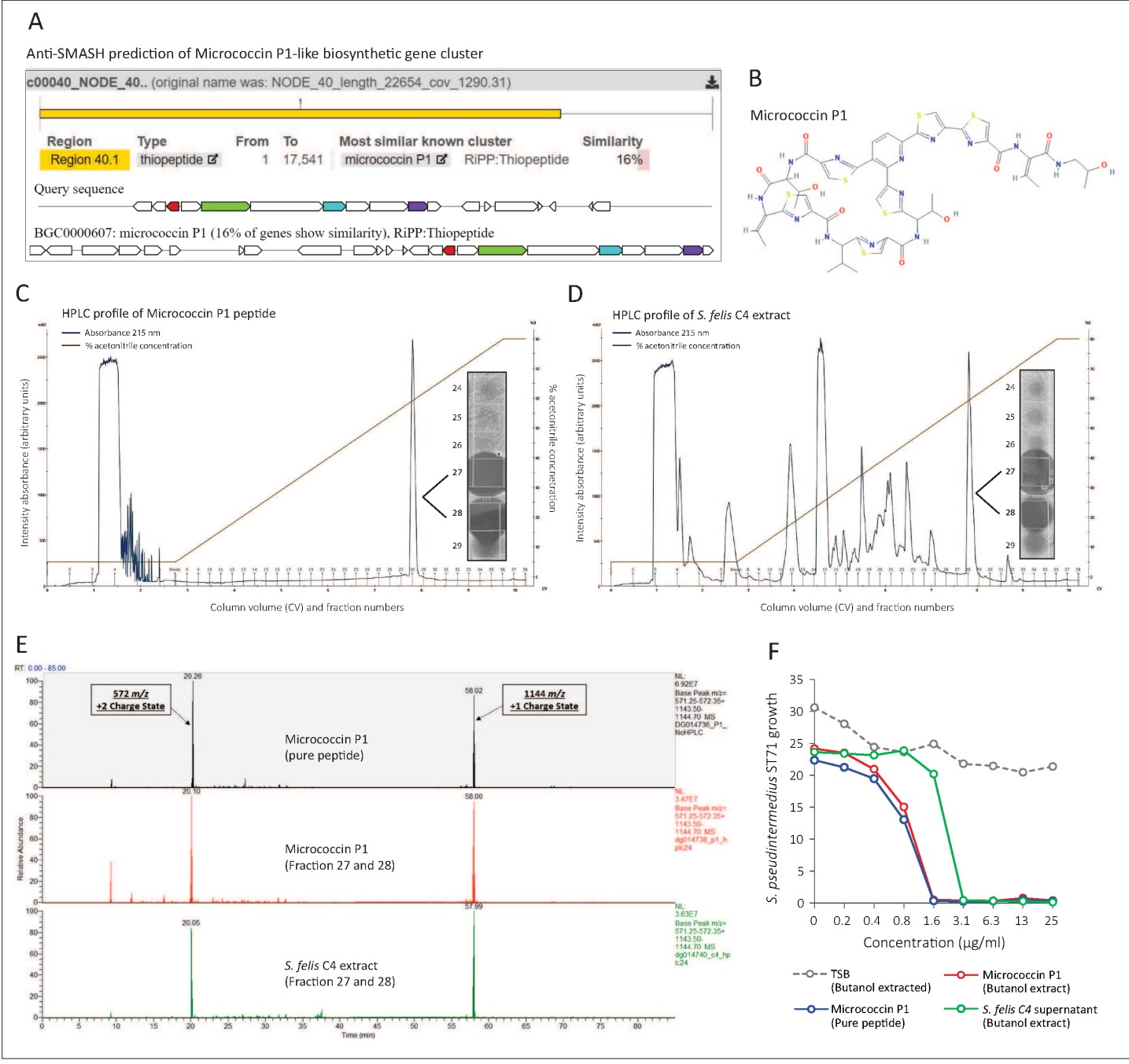

**Figure 4.** Identification of a micrococcin P1-like antimicrobial in *S. felis* C4 extract. (**A**) Prediction of a biosynthetic gene cluster (BCG) in the *S. felis* C4 genome that has predicted similarity to a micrococcin P1 thiopeptide-encoding BCG. (**B**) Micrococcin P1 chemical structure downloaded from PubChem. (**C**) HPLC chromatogram of pure micrococcin P1 showing a single major peak eluted at 59 % acetonitrile into fractions 27 and 28. Inset image shows antimicrobial activity of fractions 27 and 28 against *S. pseudintermedius* ST71. (**D**) HPLC chromatogram of *S. felis* C4 extract showing a single major peak eluted at 59 % acetonitrile into fractions 27 and 28. Inset image shows antimicrobial activity of fractions 27 and 28 against *S. pseudintermedius* ST71. (**E**) Mass spectrometry chromatogram of similar masses and charge states generated from the synthetic peptide micrococcin P1 (top panel), HPLC fractions 27 and 28 from micrococcin P1 (middle panel) and HPLC fractions 27 and 28 from *S. felis* C4 extract (bottom panel). (**F**) Antimicrobial activity of negative control TSB, *S. felis* C4 supernatant and micrococcin P1 (before or after extraction with n-butanol) against *S. pseudintermedius* ST71 (OD600 nm) after 18 hr. Representative of two independent butanol extractions and antimicrobial assays.

The online version of this article includes the following figure supplement(s) for figure 4:

**Source data 1.** Source data for *Figure 4C and D* (labelled and unlabelled images of the antimicrobial agar assay from HPLC fractions of micrococcin P1 and *S. felis* C4 extract).

**Figure supplement 1.** Spectral fingerprint of micrococcin P1 from HPLC eluted fractions.

(*Figure 1E*), we speculated that *S. felis* C4 activity may target and compromise the bacterial membrane and/or cell wall. To address this question, we utilized bacterial cytological profiling (BCP) to distinguish between the different cellular pathways targeted by antimicrobials (*Nonejuie et al., 2013*). *B. subtilis* cells exposed to 1 X MIC of the *S. felis* C4 extract and micrococcin P1 showed greater elongation compared to DMSO control and nucleoids that were highly condensed into toroidal structures, a characteristic of compounds that block translation, such as tetracycline (*Figure 5A*). Nucleoids were also highly condensed in micrococcin P1-treated cells and to a lesser extent in PSMβ2-treated cells. To examine the ultrastructural profile of staphylococci, we conducted transmission electron micros-copy (TEM) imaging on sectioned *S. pseudintermedius* ST71 cells that were exposed to sub-MIC (0.8 μg/ml), MIC (1.5 μg/ml), or 5 X MIC (25 μg/ml) amounts of *S. felis* C4 extract or DMSO control (*Figure 5B*). TEM observations upon control DMSO treatment showed normal, uniform spherical cocci physiology and septum formation indicating active replication. In contrast, short exposure to the extract showed evidence of cell wall thickening, membrane invagination, and major perturbations in the structure and rigidity of the cell membrane (*Figure 5B*, lower zoom inset panels). Moreover, treat-ment with the extract showed evidence of greater chromosomal compaction compared to control, evidenced by the increased (electron) density of the nucleoid (highlighted yellow arrows). This obser-vation is consistent with the cytological profile of toroidal structures in *B. subtilis* by BCP, indicating a similar mechanism of action in staphylococci. Additionally, exposure of *S. pseudintermedius* ST71 to increasing concentrations of the *S. felis* extract, resulted in a dose-dependent increase in reac-tive oxygen species (ROS) (*Figure 5C*) together with a concomitant decrease in the intracellular ATP levels (*Figure 5D*). Furthermore, evidence to support an additional membrane-active component was provided by flow cytometry analysis of bacterial cells positively stained with the membrane imperme-able dye propidium iodide (PI). Greater than 20 % of *S. pseudintermedius* cells were PI-positive after 4 h exposure to 1 X-4X MIC of *S. felis* extract (*Figure 5E*). Lastly, *S. felis* C4 was shown to potentiate the effects of antibiotics against *S. pseudintermedius* ST71. The MIC values of gentamicin, rifampicin, and chloramphenicol were decreased in the presence of sub inhibitory 0.25 X MIC of *S. felis* extract compared to TSB control alone (*Figure 5F*). Remarkably, growth of erythromycin-resistant *S. pseudin-termedius* ST71 in the presence of subMIC *S. felis* C4 extract restored sensitivity to erythromycin from MIC >100 μg/ml – MIC = 0.2 μg/ml.

## *S. felis* C4 inhibits *S. pseudintermedius* skin colonization and infection in mice

We next sought to investigate the potential of *S. felis* C4 and its antimicrobial products as a therapy against *S. pseudintermedius* colonization and infection on an animal model. Since *S. felis* C4 is a commensal bacterium that was isolated from healthy feline skin, we speculated it should be safe and well tolerated on mammalian skin. *S. felis* C4 was sensitive to several common antibiotics (*Figure 6A*) and as such represents a suitable strain for further investigation as a bacteriotherapy. We therefore assessed the skin tolerability of a 3 day topical application of *S. felis* C4 on SKH1 hairless mice. Whereas *S. pseudintermedius* and the non-antimicrobial *S. felis* ATCC 49168 showed evidence of disease as observed by scaling and redness on back skin, *S. felis* C4 did not promote any adverse reaction (*Figure 6B*).

To test the antimicrobial activity of *S. felis* C4 on skin, we first applied $5 \times 10^7$ CFU/cm$^2$ *S. pseud-intermedius* directly onto mouse back skin, 48 hr later we then applied an equal density of *S. felis* C4 or 1 mg of the *S. felis* C4 extract to the same site. Both treatments were repeated daily for three days. The skin showed a reduction in scaling and redness post-treatment with *S. felis* C4 or extract compared to control (*Figure 6C*). Enumeration of *S. pseudintermedius* ST71 CFUs after plating onto selective agar revealed a significant 2.9 log decrease in CFU after extract application and a 3.3 log decrease after live *S. felis* C4 application (*Figure 6D*). Enumeration of total staphylococci CFUs after plating onto selective agar confirmed that the extract treatment significantly reduced bacterial colonization (*Figure 6E*). In contrast, total staphylococci CFU counts after the *S. felis* C4 application were more similar to the control group, suggesting that the *S. felis* bacteria successfully colonized and outcompeted *S. pseudintermedius* ST71 on the skin (*Figure 6E*). To further test the *S. felis* C4 extract as an anti-MRSP intervention, we also evaluated its efficacy in limiting deep tissue infection by *S. pseudintermedius* injected into the dermis. One hour after an intradermal injection of $1 \times 10^7$ CFU *S. pseudintermedius* ST71, two intradermal inoculations of 250 μg of *S. felis* C4 extract were

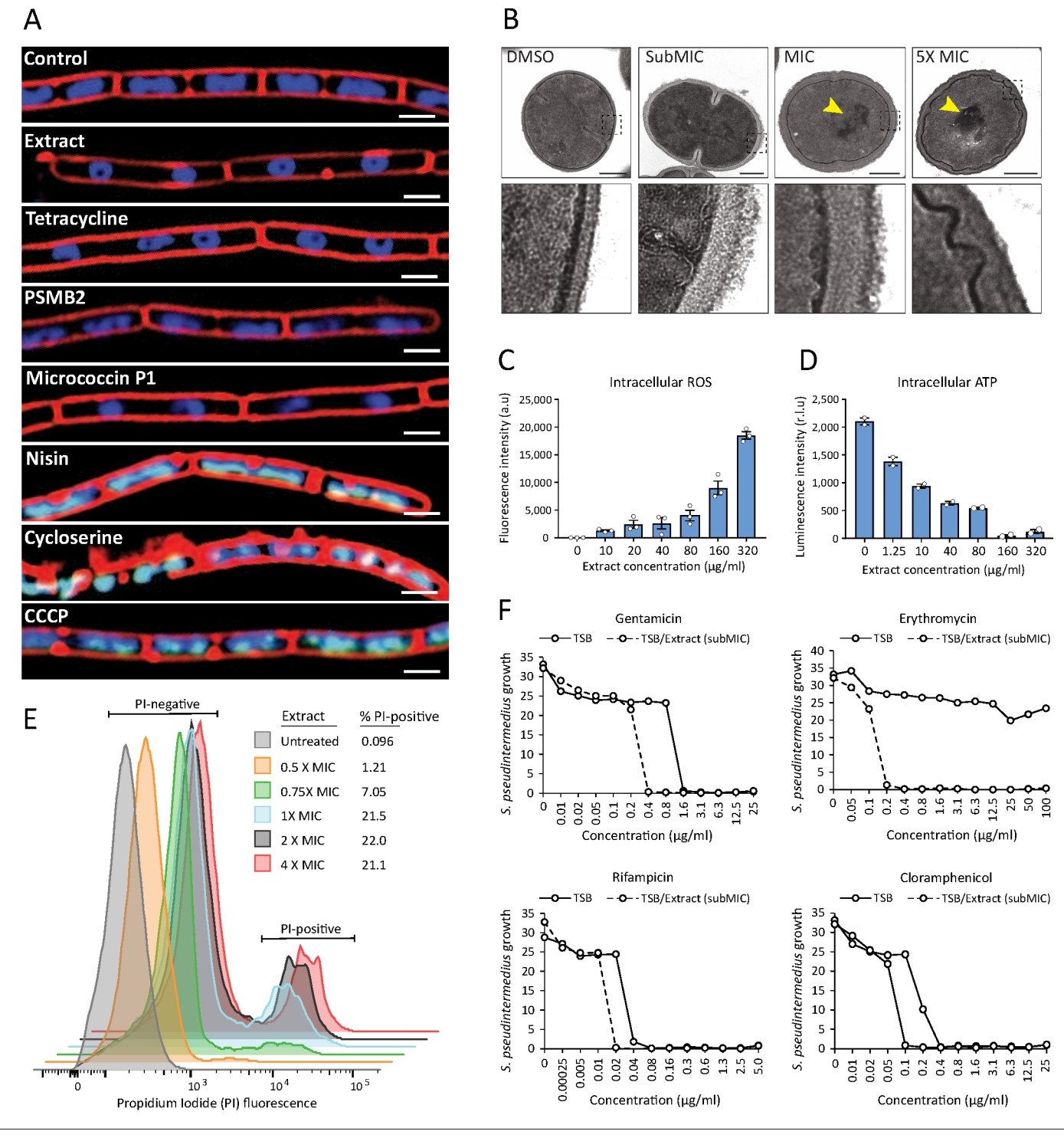

**Figure 5.** *S. felis* C4 antimicrobials target the bacterial cell membrane inhibit protein translation. (**A**) Cytological profiles of *B. subtilis* PY79 upon treatment with 1 X MIC of *S. felis* C4 extract, tetracycline, PSMβ2, micrococcin P1, nisin, cycloserine, CCCP (carbonyl cyanide m-chlorophenylhydrazine) and DMSO control. Fluorescent microscopy images were taken 2 hr post-treatment, except for cycloserine, nisin and CCCP which were at 30 m post-treatment. The cell membrane was stained red with FM4-64, DNA-stained blue with DAPI, or green with SYTOX when the integrity of the membrane was compromised. Scale bar = 2 μm. (**B**) TEM images of *S. pseudintermedius* ST71 after 1 hr treatment with *S. felis* C4 extract or DMSO control at the indicated concentrations. Yellow arrows highlight areas of condensed DNA. Lower image panels represent higher magnification of regions highlighted by dashed black boxes. Scale = 250 nm. (**C**) Total ROS accumulation in *S. pseudintermedius* ST71 after 1 hr treatment in increasing concentrations of *S. felis* C4 extract. Error bars indicate SEM. (**D**) Total intracellular ATP accumulation in *S. pseudintermedius* ST71 after 1 hr treatment in increasing

*Figure 5 continued on next page*

*Figure 5 continued*

concentrations of *S. felis* C4 extract. Error bars indicate SEM. (**C–D**) representative of two separate experiments. (**E**) Flow cytometric LIVE/DEAD viability assay and quantification of SYTO9-positive *S. pseudintermedius* ST71 that were propidium iodide-positive (PI) or PI-negative at 4 hr post-treatment with increasing concentrations of *S. felis* C4 extract. Representative of two independent experiments. (**F**) Growth of *S. pseudintermedius* ST71 (OD600 nm) in TSB or TSB supplemented with a sub inhibitory 0.25 X MIC (0.4 µg/ml) of *S. felis* C4 extract after 18 hr incubation with increasing concentrations of rifampicin, gentamicin, erythromycin or cloramphenicol. Representative of three independent experiments.

The online version of this article includes the following figure supplement(s) for figure 5:

**Source data 1.** Source data for *Figure 5A* (folder of uncropped images generated by bacterial cytological profiling of *B. subtilis* PY79).

**Source data 2.** Source data for *Figure 5B* (Uncropped EM images of *S.p* ST71).

**Source data 3.** Source data for *Figure 5C and D* (values of ROS and ATP measured by luminescence) and *Figure 5F* (growth of *S.p* ST71 in the presence of antibiotics).

administered adjacent to the infection site. Infection was monitored by observations of skin necrotic lesion size over a 14 -day period. Compared to controls, the extract-treated mice exhibited slower lesion progression from day 1 to day 2, and significantly better protection from *S. pseudintermedius* skin disease from day four onwards (*Figure 6F and G*). These results demonstrate the in vivo efficacy and clinical potential for *S. felis* C4 as a bacteriotherapy against *S. pseudintermedius* skin colonization and infection.

## Discussion

*S. pseudintermedius* is one of the most common pathogens isolated from the skin of dogs and is becoming increasingly prevalent on humans. The incidence of severe and recurrent infections in animals and humans caused by methicillin-resistant *S. pseudintermedius* is associated with a predominant clone belonging to the multi-locus sequence type ST71 (*Darlow et al., 2017*; *Perreten et al., 2010*; *Riegel et al., 2011*; *Robb et al., 2017*; *Stegmann et al., 2010*; *Weese et al., 2009*). ST71 isolates exhibit resistance to many classes of antibiotics and represent a considerable therapeutic challenge. In the current study, high-throughput antimicrobial screening of a collection of diverse animal-derived staphylococci led to the discovery of *S. felis* C4. This strain secretes antimicrobials that inhibit the growth of several drug-resistant gram-positive pathogens by disrupting the cell membrane and inhibiting protein translation. Although *S. felis* remains poorly characterized in the literature, it is the most frequent species of staphylococci isolated from cats and is susceptible to most antimicrobials (*Worthing et al., 2018b*). In this study, 23 *S. felis* isolates were screened but only three showed reproducible antimicrobial activity in liquid and agar co-culture, suggesting an uncommon intraspecies trait. Importantly, we demonstrated that topical application of the live *S. felis* C4 organism outcompeted MRSP colonization in vivo, likely by the active secretion of its antimicrobials on the skin surface. Indeed, topical application of the sterile antimicrobial extract was similarly effective in reducing CFU counts on mouse skin and significantly reduced the size of necrotic infected lesions. These positive in vivo findings build upon other reports of the utilization of commensal bacteria as biotherapeutic products to treat skin diseases.

There are now several reports of the discovery and characterisation of distinct antimicrobial-producing strains within common human bacterial skin species, including *S. epidermidis* (*Cogen et al., 2010*) *S. hominis* (*Nakatsuji et al., 2017*), *S. lugdunensis* (*Zipperer et al., 2016*), *S. capitis* (*O'Neill et al., 2020*) and recently *Cutibacterium acnes* (*Claesen et al., 2020*). Unfortunately, the frequency and abundance of such antimicrobial isolates are found to be significantly reduced during human AD (*Nakatsuji et al., 2017*). Indeed, this phenomenon of a relative absence of protective commensal strains could also influence canine AD, which shares many of the clinical features of human AD and a corresponding predisposition to *S. pseudintermedius* colonization.

In the treatment of AD, it has been reported that topical application of a 5 % lysate from the gram-negative bacterium *Vitreoscilla filiformis* can be effective in human and mouse AD models (*Gueniche et al., 2008*). Another group reported improved outcomes in mice and on humans when gram-negative bacterium from soil called *Roseomonas mucosa* was applied to skin (*Myles et al., 2016*), although subsequent controlled clinical trials have not confirmed the effectiveness of this approach. A major caveat to these studies is the use of gram-negative bacteria with unclear mechanism of action and which do not survive well on the skin surface. Isolation of potentially protective bacteria from taxa

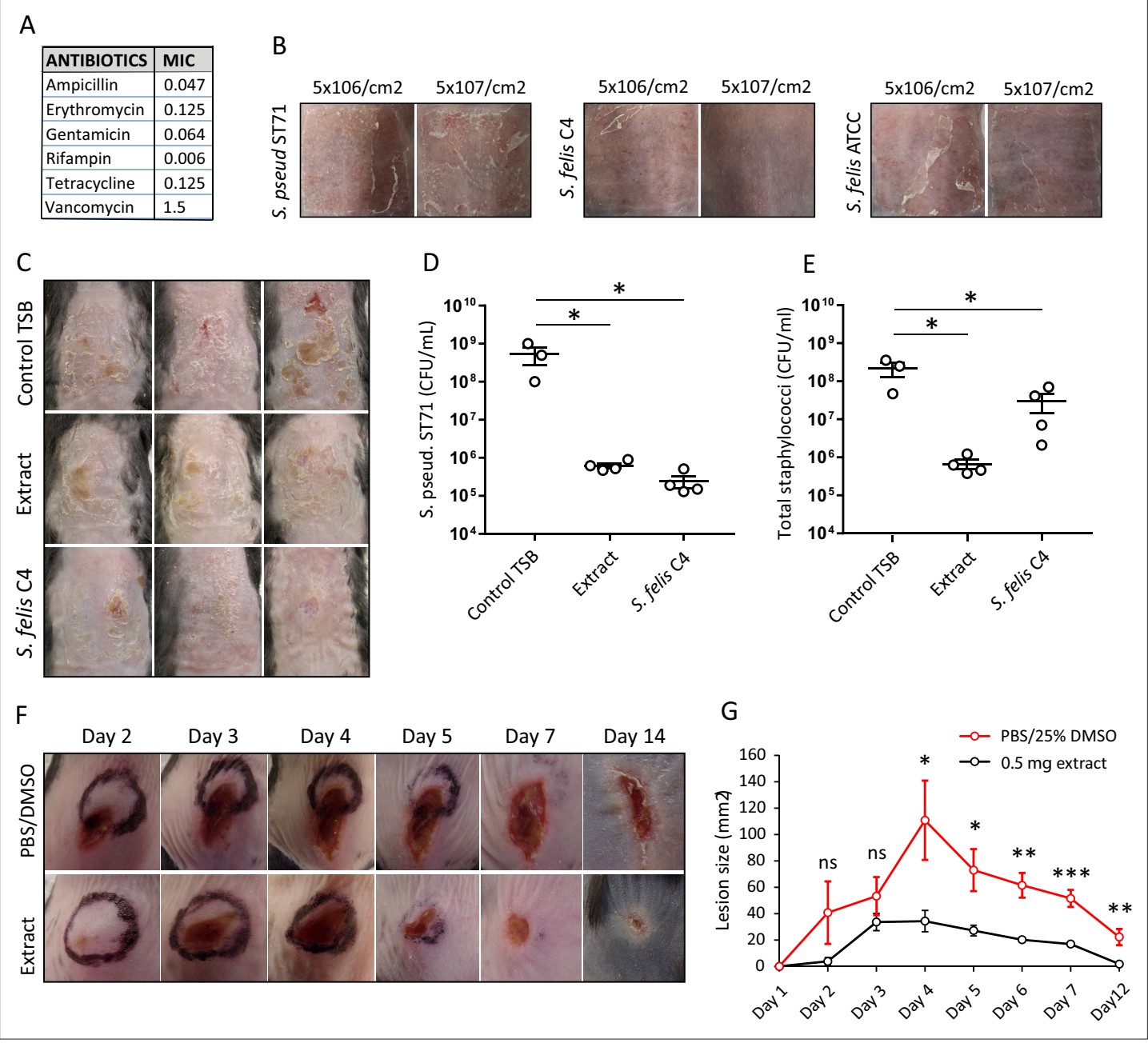

**Figure 6.** Live bacteriotherapeutic intervention with *S. felis* C4 protects against *S. pseudintermedius* colonization in mice. (**A**) Minimum inhibitory concentrations (MIC) of the indicated antibiotics against *S. felis* C4. (**B**) Representative images of the dorsal skin of 8–10 week-old SKH1 mice 3 days post-challenge with live *S. felis* C4, *S. pseudintermedius* ST71 (*S. pseud* ST71) or *S. felis* ATCC (49168), inoculated at the indicated amounts. n = 2, per treatment group. (**C–E**) 5 × 10⁷ CFU/cm² of *S. pseudintermedius* ST71 was applied onto the back skin of C57BL/6 mice for 48 hr and challenged with TSB, *S. felis* C4 extract (1 mg) or live *S. felis* C4 (5 × 10⁷ CFU/cm²) for 72 hr. Post-treatment, mouse back skin was photographed (**C**) and swabbed to enumerate *S. pseudintermedius* ST71 CFU on selective Baird-Parker egg yolk tellurite agar (**D**) or total staphylococci CFU on selective mannitol-salt agar plates (**E**). n = 3 for TSB and n = 4 for extract and *S. felis* C4. Error bars indicate SEM. One-way ANOVA with multiple comparisons (Dunnett's correction) was performed. p values: *p < 0.05; (**F–G**) At day 0, 1 × 10⁷ CFU of *S. pseudintermedius* ST71 was intradermally injected into the back skin of 8–10 week-old C57BL/6 mice and at 1 hr post-infection two inoculations of *S. felis* C4 extract (250 µg) or PBS/25 % DMSO control were injected adjacent to the infection site. (**F**) Representative images of *S. pseudintermedius* ST71-induced dermonecrosis over time after receiving control PBS/DMSO or *S. felis* C4 extract. (**G**) Quantification of lesion size (mm²) over time as measured by L x W of lesions. n = 4 for DMSO/PBS and n = 5 for extract. Error bars indicate SEM. A two-tailed, unpaired Student's *t*-test was performed. p values: *p < 0.05; **p < 0.01; ***p < 0.001.

The online version of this article includes the following figure supplement(s) for figure 6:

**Source data 1.** Source data for *Figure 6D and E* (CFU counts on mouse back skin for *S.p* ST71 and total CoNS) and *Figure 6G* (measurements of lesion sizes over time in *S. p* ST71 infected skin).

that have evolved the capacity to survive on the skin will maximize the capacity to attack pathogen targets. For example, in a randomized, double-blind, placebo-controlled trial *S. aureus* colonization of AD patient skin was significantly reduced after application of a human commensal *S. hominis* strain (*Nakatsuji et al., 2021b*; *Nakatsuji et al., 2021a*). This anti-*S. aureus* activity was mediated by several secreted antibiotics unique to *S. hominis* A9. The current work suggests *S. felis* C4 also produces several antimicrobial peptides that could be useful in animal or human applications.

Mass spectrometry analysis of the semi-purified *S. felis* C4 extract identified several peptide antimicrobials including amphipathic, α-helical PSMβ and a thiopeptide similar to micrococcin P1. Thiopeptides constitute a family of ribosomally synthesized and posttranslationally modified peptides (RiPPs) (*Ciufolini and Lefranc, 2010*). The BGC encoding micrococcin P1 has been previously reported to reside on a 24 kb plasmid (*Bennallack et al., 2014*). Likewise, the BCG for micrococcin P1 resided on a 23 kb contig in *S. felis* C4 with several plasmid associated genes present. Micrococcin P1 is known to be active predominantly against gram-positive bacteria, targeting the L11 binding domain of the 23 S ribosomal RNA, thereby inhibiting translation (*Chan and Burrows, 2020*). A similar mode of action was observed from the cytological and ultrastructural profiles of bacteria exposed to *S. felis* C4 extract, resulting in condensed nuclei and toroid shaped chromosomes, indicative of stalled translation. The multi-step, complex biosynthesis of micrococcin P1 and its relatively poor physicochemical properties have hampered its development into an effective anti-infective. As a result other groups have sought to develop a micrococcin P1 nanoparticle formula to circumvent its poor pharmacokinetics (*Liu et al., 2020*). Another group has recently reported the total laboratory synthesis of micrococcin P1 through a scalable thiazole forming sequence that produced synthetic micrococcin that was spectroscopically and functionally indistinguishable from the natural product (*Christy et al., 2020*). Here, we show that micrococcin P1 can also be extracted from bacterial culture supernatant using n-butanol. This extract was not found to be cytotoxic at concentrations that were antimicrobial and effectively inhibited growth of MRSP in vivo by direct topical application. The antimicrobial activity, antibiotic sensitivity and tolerability of *S. felis* C4 suggests it may be suitable for bacteriotherapy.

Small α-helical AMPs, which include mammalian cathelicidin (active LL-37) (*Nakatsuji and Gallo, 2012*) as well as bacterial PSMs (*Cogen et al., 2010*; *Zeng et al., 2019*), are also attractive candidates as anti-infectives. The family of phenol-soluble modulins which include PSMα, PSMβ, PSMγ, and PSMε, are a class of small, immunomodulatory AMPs found in many staphylococci species. PSMα and PSMγ are classic cytolytic toxins but PSMβ do not exhibit cytotoxic activity against eukaryotic membranes despite a strong affinity to bind and lyse POPC vesicles, which mimic biological membranes (*Duong et al., 2012*). As a result, the biological function of PSMβ has remained unclear (*Cheung et al., 2014*). However, several recent studies have reported the antimicrobial activity of PSMβ (*Kumar et al., 2017*; *O'Neill et al., 2020*). Here, MS detected several PSMβ peptides that were highly enriched in an antimicrobial HPLC fraction from *S. felis* C4 supernatant. Due to their identical size, attempts to purify the individual PSMs by HPLC were unsuccessful. Instead, their activity against MRSP was validated using synthetic versions of the peptides, but they exhibited less potency than the extract. One potential explanation is that compared to the smaller PSMs α and γ (20–25 amino acids), that consist of a single α-helix, PSMβ are much larger (43–44 amino acids) and contain three α-helices. Solution structures of PSMβ from *S. aureus* revealed that two of the α-helices fold to form a 'V-shape', producing a hydrophobic core that is similar to other bacteriocins (*Towle et al., 2016*). As such, it remains plausible that the synthetic PSMβ peptide versions did not fold correctly and that this negatively affected their antibacterial activity. Future research efforts will attempt to develop and characterize truncated and mutated versions of *S. felis* PSMs with the goal of enhancing antimicrobial activity for a more simplistic but potentially more powerful therapeutic (*Zeng et al., 2019*). Nevertheless, both fluorescence and electron microscopy of bacteria exposed to the extract showed drastic perturbations of the bacterial cell membrane and cell wall thickening, which is consistent with the membrane-targeting actions of amphipathic AMPs. The concomitant accumulation of bacterial ROS and decrease in ATP production are also consistent with bacterial membrane disruption and increased permeability (*Song et al., 2020*). Overall, the ultrastructural response and killing of bacteria in response to *S. felis* C4 suggests both PSMs and micrococcin made by this strain are active and produce a highly potent antimicrobial commensal microbe that helps protect the host from gram-positive pathogens.

In addition to their protection against pathogen colonization, skin commensals play important roles in promoting skin health and immune homeostasis. Although PSMs are common amongst

staphylococci, pathogenic *S. aureus* exhibits a preference for PSMα production over PSMβ. In contrast, commensal staphylococci production of PSMβ is prioritized over the more toxic PSMα and PSMγ versions, a feature suspected to be an evolutionary adaptation to stably colonize skin (*Otto, 2009*; *Wang et al., 2007*). Naturally, *S. felis* C4 PSMβ and extract treatment of NHEKs yielded minimal evidence of cytotoxicity, whereas PSMγ induced extensive cytotoxicity. These smaller PSMs are well characterised toxins that trigger pro-inflammatory responses (*Nakamura et al., 2013*; *Williams et al., 2019*). Whereas the larger PSMβ reportedly does not elicit pro-inflammatory activity in vitro – a finding that was supported by our data, little is known regarding other potential host responses to PSMβ exposure (*Cheung et al., 2014*). We speculated whether PSMβ might exhibit anti-inflammatory activities, and in the present context that activity would be therapeutically beneficial. Indeed, when both NHEKs and THP-1 macrophages were treated with *S. felis* PSMs or extract in the presence of TLR agonists, cytokine induction was reduced but most evidently by PSMβ2. RNA-Seq analysis of NHEKs revealed global suppression of inflammatory pathways typically activated by TLR3, in the presence of PSMβ2.

Taken together, here we report that *S. felis* C4 has low cytotoxicity and broad-spectrum antimicrobial activity and anti-inflammatory activity and is therefore an attractive biotherapeutic candidate for treatment of skin disease.

# Materials and methods

## Key resources table

| Reagent type (species) or resource | Designation | Source or reference | Identifiers | Additional information |
|---|---|---|---|---|
| Strain, strain background (*Escherichia coli*) | K1(RS218) | PMID:26205862 | | Provided by the Nizet lab |
| Strain, strain background (*Acinetobacter baumannii*) | AB 5075 | PMID:24865555 | | Provided by the Nizet lab |
| Strain, strain background (*Pseudomonas aeruginosa*) | PAO1 | PMID:10984043 | | This study |
| Strain, strain background (*Staphylo coccus aureus*) | Newman, USA300,113, 6538, 33,591 | ATCC, PMID:17951380 | | This study |
| Strain, strain background (*Staphylo coccus aureus*) | NIAMS | PMID:28228596 | Lesional skin of atopic dermatitis patients | |
| Strain, strain background (*Enterococcus faecium*) | VRE 5998 | | | Provided by Nizet lab |
| Strain, strain background (*Klebsiella pneumoniae*) | K1100 | PMID:31353294 | | Provided by Nizet lab |
| Strain, strain background (*Bacillus subtilis*) | PY79 | PMID:24356846 | | This study |
| Cell line (*Homo-sapiens*) | (NHEK) normal human epidermal keratinocytes | Thermo Fisher | C0015C | |
| Antibody | Anti-Phospho-TBK1/NAK (Ser172) (Rabbit mAb) | Cell Signaling | Cat# D52C2 | WB (1:1,000) |
| Antibody | Anti-TBK1/NAK (Rabbit mAb) | Cell Signaling | Cat# D1B4 | WB (1:1,000) |
| Antibody | Anti-Phospho-IRF-3 (Ser396) (Rabbit mAb) | Cell Signaling | Cat# 4D4G | WB (1:1,000) |
| Antibody | Anti-IRF-3 (Rabbit mAb) | Cell Signaling | Cat# D83B9 | WB (1:1,000) |
| Antibody | Anti-COX IV (Rabbit mAb) | Cell Signaling | Cat# 3E11 | WB (1:1,000) |
| Antibody | LICOR IRDye 680RD Donkey anti-Goat IgG | Licor | Cat# LIC-926–68074 | WB (1:10,000) |
| Antibody | LICOR IRDye 800CW Donkey anti-Rabbit IgG | Licor | Cat# LIC-925–32213 | WB (1:10,000) |

*Continued on next page*

*Continued*

| Reagent type (species) or resource | Designation | Source or reference | Identifiers | Additional information |
|---|---|---|---|---|
| Sequence-based reagent | DEFB4A | Integrated DNA Technologies | PrimeTime predesigned qPCR primers | Assay ID# Hs.PT.58.40718840 |
| Sequence-based reagent | GAPDH | Integrated DNA Technologies | PrimeTime predesigned qPCR primers | Assay ID# Hs.PT.39a.22214836 |
| Sequence-based reagent | IL-6 | Integrated DNA Technologies | PrimeTime predesigned qPCR primers | Assay ID# Hs.PT.58.40226675 |
| Sequence-based reagent | CXCL10 | Integrated DNA Technologies | PrimeTime predesigned qPCR primers | Assay ID# Hs.PT.58.3790956.g |
| Commercial assay or kit | Thermo Pierce Silver stain kit | Thermo Fisher | Cat. #: 24,612 | |
| Commercial assay or kit | iTaq universal SYBR green supermix | Biorad | Cat. #: 1725121 | |
| Commercial assay or kit | Pierce BCA Protein Assay Kit | Thermo Fisher | Cat. #: 23,225 | |
| Commercial assay or kit | Pierce LDH Cytotoxicity Assay Kit | Thermo Scientific | Cat. #: PI88954 | |
| Commercial assay or kit | Human IP-10 ELISA Set | BD Biosciences | Cat. #: 550,926 | |
| Commercial assay or kit | PureLink RNA Mini Kit | Thermo Fisher | Cat. #: 12183025 | |
| Commercial assay or kit | LIVE/DEAD *Bac*Light Bacterial Viability Kit | Thermo Fisher | Cat. #: L7012 | |
| Commercial assay or kit | 2 ml 96-well sterile deepWell plates | Fisher Scientific | Cat. #: 12-565-605 | |
| Chemical compound, drug | n-butanol 99.9% | Sigma Aldrich | Cat. #: 537,993 | |
| Chemical compound, drug | Crystal violet solution (1%) | Sigma Aldrich | Cat. #: V5265 | |
| Chemical compound, drug | Micrococcin P1 | Cayman Chemical | Cat. #: 17,093 | |
| Chemical compound, drug | Poly I:C | InvivoGen | Cat. #: tlrl-pic | |
| Chemical compound, drug | MALP-2 | Enzo Life Sciences | Cat. #: ALX-162–027 C050 | |
| Chemical compound, drug | Precision Plus Protein Dual Xtra Prestained Protein Standard | Biorad | Cat. #: 1610377 | |
| Chemical compound, drug | Human keratinocyte growth supplement | Thermo Fisher | Cat. #: S0015 | |
| Chemical compound, drug | Trypsin/EDTA solution (TE) | Thermo Fisher | Cat. #: R001100 | |
| Chemical compound, drug | Defined trypsin inhibitor | Thermo Fisher | Cat. #: R007100 | |
| Chemical compound, drug | EpiLife medium with 60 µM calcium | Thermo Fisher | Cat. #: MEPI500CA | |
| Chemical compound, drug | Antibiotic Antimycotic Solution (100 X) | Millipore Sigma | Cat. #: A5955 | |
| Chemical compound, drug | RIPA Lysis and Extraction Buffer | Fisher Scientific | Cat. #: PI89900 | |
| Chemical compound, drug | Halt Protease and Phosphatase Inhibitor Cocktail (100 X) | Thermo Fisher | Cat. #: 78,440 | |

*Continued on next page*

*Continued*

| Reagent type (species) or resource | Designation | Source or reference | Identifiers | Additional information |
|---|---|---|---|---|
| Chemical compound, drug | Intercept (PBS) Blocking Buffer | Licor | Cat. #: 927–70001 | |
| Software, algorithm | GraphPad Prism 7.03 | GraphPad Software Inc | | |
| Software, algorithm | FlowJo V10 | BD Biosciences | | |

## Bacterial strains and growth conditions

The bacterial strains used in this study were all grown overnight, with the exception of *E. faecium* which was grown for 48 hr, in 3 % Tryptic Soy Broth (TSB) with shaking at 250 rpm in a 37 °C incubator or otherwise grown on agar at 37 °C under static conditions.

## Sample collection

Animal-derived staphylococci samples came from two previously described collections: the first collection consisted of clinical isolates of skin and soft tissue infection from Australian dogs and cats (*Worthing et al., 2018a*), and the second collection was comprised of staphylococci isolated from the nose, mouth and perineum of healthy dogs and cats in Australia (*Ma et al., 2020*). All samples had previously been identified by matrix assisted laser desorption ionization-time of flight mass spectrometry (MALDI-TOF), as previously described (*Worthing et al., 2018b*), and the ST71 MRSP isolate had been characterized by whole genome sequencing (*Worthing et al., 2018a*). Two human derived antimicrobial skin commensal isolates were used as positive controls: *S. hominis* A9 (*Nakatsuji et al., 2017*) and *S. capitis* E12 (*O'Neill et al., 2020*) and a non-antimicrobial *S. aureus* 113 isolate served as a negative control.

## In vitro antimicrobial screen

For the initial staphylococci screen, single clone-derived cultures of animal-derived staphylococci were used as competitor isolates against the growth of methicillin-resistant *S. pseudintermedius* ST71. Each pure culture, including positive and negative control strains, were first streaked onto 3 % TSB agar plates and a single colony was transferred to 1 ml of TSB in a deep 96 well plate (Thermo). The CoNS plate was sealed with sterile Aeraseal film (Sigma, St. Louis, MO) and cultured at 37 °C overnight with shaking at 250 rpm. Bacterial growth was evaluated by measuring OD600 with only bacteria that grew to a density (OD600 >6.0) used for subsequent analysis. To measure antimicrobial activity in the secreted supernatant, the animal-derived staphylococci supernatant from overnight cultures were harvested and centrifuged through a 96-well 0.2 μm sterile filter plate (Corning). Next, $1 \times 10^5$ CFU of *S. pseudintermedius* ST71 was inoculated into 150 μl of TSB media containing 50 %, 20% or 12.5 % sterile supernatant and grown on a plate shaker for 18 h at 30 °C. To measure antimicrobial activity from the live agar co-culture assays (radial diffusion), 20 μl of overnight *S. pseudintermedius* ST71 culture was first inoculated into 45 °C molten TSB and immobilized after pouring and cooling into square petri dishes with grids. Overnight cultures of animal-derived staphylococci were centrifuged to pellet the bacteria, washed 2 X with PBS and resuspended in fresh TSB. The culture (10 μl) was inoculated onto a 13 mm grid of the *S. pseudintermedius* agar plates and cultured overnight at 30 °C. The resulting zones of inhibition from antimicrobial isolates were imaged using the camera feature on an iPhone 12.

## Extraction and purification of antimicrobials from bacterial supernatant

Supernatant from overnight cultures of selected human and animal-derived staphylococci were first sterilized by filtration through a low-protein binding 0.22 μm Millipore filter. Activity was precipitated by ammonium sulfate (75 % saturation) for 1 hr, under constant rotation followed by centrifugation at 4,000 x g for 45 min and re-suspension of the pellet in dH$_2$O. To test stability, the precipitate was boiled at 95 °C for 30 m or stored in a sterile eppendorf tube at room temperate for 1 week. Antimicrobial activity was measured by radial diffusion assay. Sterile supernatant of S. *felis* strains were also subject to n-butanol extraction and purification as previously described (*Joo and Otto, 2014*). Briefly,

in each tube 10 ml of butanol was added to 30 ml of supernatant and incubated at 37 °C for 2 h under constant rotation. The tubes were then set aside for several mins until the butanol phase settled. After centrifugation at 2000 x g for 5 min, the upper butanol phase was collected and lyophilized in a SpeedVac vacuum concentrator. The lyophilized extract was resuspended and concentrated to 10 mg/ml in DMSO. Protein concentration was determined by Pierce BCA Protein Assay Kit.

## Determination of minimum inhibitory concentration (MIC)

MIC values were determined using a broth micro dilution method. Bacterial cells were grown to mid-late log phase, to an OD600 nm value of roughly 1.0 for each bacterial strain and then normalized to $1 \times 10^7$ CFU/mL. The PSM peptides or butanol extracts were dissolved in DMSO to a stock concentration of 10 mg/ml. The stock concentrations of antibiotics that were water-soluble were prepared with $H_2O$ or 100 % ethanol if water-insoluble. The $1 \times 10^7$ CFU/ml bacterial cultures (10 µl) were aliquoted into 96-well microtiter plates and mixed with 95 µL of media with or without twofold dilutions of the conditioned supernatant, PSM peptides, butanol extracts or antibiotics and incubated for 16–18 hr at 30 °C with shaking at 250 rpm. Growth inhibition was determined by measuring the OD600 nm readings of each well using a microplate reader (SpectraMax iD3, Molecular Devices). The MIC of each bacterial strain was determined by the lowest peptide concentration that inhibited more than 80 % bacterial growth.

## Crystal violet staining for biofilm disruption

Overnight culture of *S. pseudintermedius* ST71 was diluted in fresh TSB to $1 \times 10^7$ CFU/ml by OD600 reading. A total of 100 µl of bacteria was transferred to a flat-bottom 96 well plate and incubated at 37 °C without shaking for 4 hr to initiate biofilm formation. Next, the supernatants were removed by washing the plates three times with 200 µl of $dH_2O$. Subsequently, 150 µl of the *S. felis* C4 supernatant, sterile extract or TSB control was added at various concentrations to the biofilm for periods between 2 and 24 hr. After incubation, the supernatant was gently removed, and the biofilm was gently washed three times with $dH_2O$ followed by air drying. Next, 150 µl of 0.1 % crystal violet (CV) solution was added to all wells containing biofilm. After 20 min of incubation with CV dye, the excess CV was removed and each well was gently washed twice with $dH_2O$. Fixed CV dye was released from the biofilm by 70 % ethanol, and absorbance was measured at 595 nm.

## Bacterial cytological profiling (BCP)

Prior to BCP, MIC of all the antimicrobial compounds to be tested were generated first. MICs for *B. subtilis* PY79 were conducted in 96-well plates. Cultures were taken from glycerol stocks and plated on LB plates for 24 hr at 30 °C. On the day of the experiment, single colonies were transferred into 3 mL liquid LB media and rolled until they reached early exponential phase (OD600 0.12–0.15). In 96-well plates, antibiotics were serially diluted down twofold across the plate. One µl of cells was added to 100 µL of LB+ antibiotic. Plates were incubated for 24 hr in a 30 °C plate shaker. Plates were then read in plate reader using an OD 600 nm spectrophotometer at T0 and T24. MIC was determined by the concentration of antibiotic at which the T24 OD600 value was 10 % or less than the control cell density. BCP was performed as described previously (*Lamsa et al., 2012*; *Nonejuie et al., 2013*; *Peters et al., 2018*). Briefly, early exponential phase *B. subtilis* PY79 was incubated with 0.5 x, 1 x, or 5 x MIC concentrations of antibiotics and rolled at 30 °C for 2 hr. Samples were taken at 30 m and 2 hr, and then dye mix was added to each for each strain. The dye mix contained 30 µg mL−1 FM 4–64, 20 µg mL−1 DAPI, and 2.5 µM Sytox Green in 1× Tbase. Six µL of cells were mixed with 1.5 µL of dye mix, and then 6 µL of that mixture was added to agarose pads (1 % agarose, 20 % LB). Cells were then imaged on an Applied Precision DV Elite optical sectioning microscope with a Photometrics Cool-SNAP-HQ2 camera, and images were deconvolved using SoftWoRx v5.5.1. Deconvolved images were then converted into TIFFs using Fiji, and then adjusted for clarity in Photoshop, producing the final images.

## HPLC purification and peptide synthesis

First step HPLC purification was carried out with 1 mg of *S. felis* C4 supernatant loaded onto a Capcell Pak C8 column (5 mm, 300 A°, 4.6 mm 250 mm) (Shiseido, Tokyo, Japan) using a linear acetonitrile gradient from 10% to 60% in 0.1 % (v/v) trifluoroacetic acid (TFA) at a flow rate of 1.0 ml/min. The

resulting fractions were lyophilized, then resuspended in water, and antimicrobial activity assessed by liquid culture assay. Up to five sequential purifications were carried out with each antimicrobial fraction pooled together for the second HPLC purification. A linear gradient of acetonitrile from 25% to 50% was used for the second purification. The eluted fractions were tested for antimicrobial activity by liquid culture and agar radial diffusion assays and the active and nonactive fractions were submitted for mass spectrometry analysis. For the purification of micrococcin P1 peptide and *S. felis* extract by HPLC, 200 µg of micrococcin P1 or 1 mg of *S. felis* C4 extract were loaded onto the C18 column using a linear acetonitrile gradient from 5% to 80% in 0.1 % TFA. The active fractions were pooled and submitted for mass spectrometry analysis to identify the presence of micrococcin P1 in the *S. felis* C4 extract. Synthetic peptides were synthesized with or without N-terminal formulation to at least 80 % purity by commercial vendors (LifeTein LLC, Somerset, NJ) and (Biomatik LLC, Delaware (PSMβ1)). Sequences of the peptides are as follows:

> PSMβ1: Formyl-MSGLIDAIKTTVEAGLNGEWADMGLGIAEIVAKGIEAISGFFG
> PSMβ2: Formyl- MSDLINAIKTTVEAGLNGEWTDMGFGIADIVAKGIDVILGFFG
> PSMβ2: Non-Formyl- MSDLINAIKTTVEAGLNGEWTDMGFGIADIVAKGIDVILGFFG
> PSMβ3: Formyl- MADLINAIKTTVEAGLNGEWTDMGFGIADIVAKGIDVISGFFG
> PSMγ: Formyl- MAADIISTIGDLVKWIIDTVNKFKK
> EF-HAND:     Non-Formyl-     MSKLTRVIVTSIMTVGFLTATLGLTAGNADAKLEGNGTLSQKQYQRLASQQF

## Silver stain and acetone precipitation of antimicrobial fractions

Twenty µg of protein from sources including the antimicrobial HPLC fractions, butanol extracts and crude supernatant were loaded onto a Novex 16 % Tricine gel and subjected to SDS-PAGE. Silver staining and de-staining of the protein gels were performed according to the manufacturer's instructions (Thermo Pierce Silver Stain Kit). A previously published protocol for acetone extraction of AMPs from SDS gels was used (*Burgess, 2009*). Briefly, a sterile razor blade was used to excise gel slices according to protein size. The gel slices were cut into small pieces and immersed in $dH_2O$ for 4 hr, with regular vortexing to elute proteins. The eluted protein was mixed with four volumes of ice-cold acetone for 1 hr at –20 °C. The samples were centrifuged at 16,000 x g for 15 min at 4 °C. The supernatant was removed and lyophilized (acetone-soluble fraction) and the resulting pellet air dried briefly and resuspended in $dH_2O$ (acetone-insoluble fraction). The antimicrobial activity of both fractions was tested by radial diffusion agar assay against *S. pseudintermedius* ST71.

## NHEK culture

NHEKs (ThermoFisher) were cultured in EpiLife medium containing 60 µM $CaCl_2$ (ThermoFisher) supplemented with 1 X human keratinocyte growth supplement (ThermoFisher) and 1 X Antibiotic Antimycotic (Millipore Sigma) at 37 °C, 5 % $CO_2$. All experiments performed on NHEKs were between passages 3 and 5 with cells at 70–80% confluency. For synthetic PSM treatments, the peptides (10–1000 µg/ml) were added to the NHEKs for 4 hr or 24 hr in dimethyl sulfoxide (DMSO).

## Immunoblot

NHEK cells were treated with Poly I:C (0.4 µg/ml), PSMβ2 (10 µg/ml) or Poly I:C and PSMβ2 combined for 15 min, 30 min, or 45 min and cells lysed in complete RIPA buffer supplemented with 1 X protease and phosphatase inhibitor cocktail (ThermoFisher). The lysate was centrifuged at 4 °C, at 13,000 rpm for 20 min and the total cytoplasmic supernatant fraction was kept at –80 °C, until future use. The total protein amount was quantified for each treatment using the Pierce BCA Protein Assay Kit according to manufacturer's instructions. Fifteen µg of total protein was reduced with β-mercaptoethanol, boiled for 5 min loaded onto a 4–20% Mini-PROTEAN TGX gel (Bio-Rad). After electrophoresis, the gel was transferred onto a polyvinylidene difluoride (PVDF) membrane. The membrane was blocked for 1 hr in Intercept (PBS) Blocking Buffer (Licor) then incubated and probed overnight at 4 °C with the following primary antibodies: P-TBK1/NAK (D52C2), TBK1/NAK (D1B4), P-IRF-3 (S396), IRF-3 (D83B9), COX IV (3E11) in Intercept Blocking Buffer supplemented with 0.05 % Tween 20. The membranes were washed for 3 X for 10 min in PBS/Tween and incubated with IRDye conjugated anti-rabbit and anti-mouse secondary antibodies (IRDye800CW and IRDye680RD; Licor, USA) in Intercept Blocking Buffer/Tween

for 1 hr at room temperature, followed by 4 X washing in PBS/Tween. Images of the membranes were acquired on an Odyssey DLx Imaging System (Licor, USA).

## Mass spectrometry

Four fractions of interest (ranging from 10 to 20 μg/mL) were dried under vacuum and resuspended in 15 μL of 5 % acetonitrile with 5 % formic acid. Next, individual LC-MS experiments were conducted on 6 μL of each sample through 85 min of data acquisition on an Orbitrap Fusion (Thermo Fisher Scientific) mass spectrometer with an in-line Easy-nLC 1000 (Thermo Fisher Scientific). A home-pulled and packed 30 cm column was triple-packed with 0.5 cm, 0.5 cm and 30 cm of 5 μm C4, 3 μm C18, and 1.8 μm C18 respectively and heated to 60 °C for use as the analytical column. Peptides were first loaded at 500 bar which was followed by a chromatography gradient ranging from 6% to 25% aceto-nitrile over 70 min followed by a 5 min gradient to 100 % acetonitrile, which was held for 10 min. Elec-trospray ionization was performed by applying 2000 V through a stainless-steel T-junction connecting the analytical column and Easy-nLC system. Each sample was followed by four washes starting with a gradient from 3% to 100% acetonitrile over 15 min with an additional 10 min at 100 % acetonitrile. An m/z range of 375–1500 was scanned for peptides with charge states between 2 and 6. Centroided data was used for quantitation of peaks. Acquisition was run in a data-dependent positive ion mode. Raw spectra were searched in Proteome Discoverer Version 2.1 and PEAKS Studio X (*Ma et al., 2003*) against 6-frame translated databases based of a uniprot reference database for *Staphylococcus felis* ATCC 49168 (Uniprot proteome UP000243559, accessed 06/26/2019) as well as in-house sequencing of *S. felis* C4 and de-novo algorithm of PEAKS Studio X. Data were searched using the Sequest algorithm (*Eng et al., 1994*) using a reverse database approach to control peptide and protein false discoveries to 1 % (*Elias and Gygi, 2007*). No enzyme was specified in the search and a minimum peptide length was set to six amino acids. Search parameters included a precursor mass tolerance of 50 ppm and fragment mass tolerance of 0.6 Da and variable oxidation for modifications.

## Whole genome sequencing and analysis of the *S. felis* C4 strain

DNA was extracted from *S. felis* C4 using the UltraClean Microbial DNA Isolation Kit (MoBio) according to the manufacturer's instructions. The library was prepared using Nextera DNA Flex library prepara-tion kit according to the manufacturer's instructions (Illumina, San Diego, CA). The library was diluted to 1.0 nM, then sequenced for 300 cycles using the Illumina NovaSeq system to generate 150 bp paired-end reads with 794 x coverage that was reduced to 100 x coverage for read mapping. Fastq files from *S. felis* C4 were trimmed using Trimmomatic (*Bolger et al., 2014*), then assembled using SPAdes Genome Assembler v.3.14.1. The *S. felis* C4 genome was annotated using the RAST tool kit via the Pathosystems Resource Integration Center (PATRIC) database (*Wattam et al., 2014*). Genes encoding PSMβ were identified by BLASTn and secondary metabolite biosynthesis gene clusters by anti-SMASH bacterial version (*Weber et al., 2015*).

## ATP determination

The intracellular ATP levels of *S. pseudintermedius* ST71 treated with *S. felis* C4 butanol extract were measured following the manufacturer's instructions (ReadiUse Rapid Luminometric ATP Assay Kit). Briefly, an overnight *S. pseudintermedius* ST71 culture was sub-cultured to an OD of 0.5 at 37 °C. The bacteria were pelleted, washed with fresh TSB and incubated with various concentrations (0–320 μg/ml) of *S. felis* C4 butanol extract for 1 hr. The bacterial cultures were centrifuged at 10,000 x g for 5 min at 4 °C. The bacterial pellet was lysed by lysozyme and centrifuged. The bacterial supernatant was mixed with an equal volume of detecting solution in a 96-well plate and incubated at room temperature for 20 min. ATP luminescence was read using a SpectraMax iD3 (Molecular Devices).

## Reactive oxygen species (ROS) measurement

The levels of reactive oxygen species (ROS) in *S. pseudintermedius* ST71 that was treated with *S. felis* C4 butanol extract were measured with 2′,7′-dichlorofluorescein diacetate (DCFDA) following the manufacturer's instructions (DCFDA/H2DCFDA Abcam cellular ROS assay kit). Briefly, an overnight culture of *S. pseudintermedius* ST71 was sub-cultured to an OD of 0.5 at 37 °C. The bacteria were pelleted and re-suspended in fresh TSB. DCFDA was added to a final concentration of 20 μM to the bacterial culture incubated with various concentrations of extract (0–320 μg/ml) at 37 °C for 1 hr.

Fluorescence intensity was immediately measured at an excitation wavelength of 488 nm and an emission wavelength of 525 nm using a SpectraMax iD3 (Molecular Devices).

## Bacterial viability assay

Dead or damaged bacteria induced by *S. felis* C4 extract were evaluated using the LIVE/DEAD *Bac*Light Bacterial Viability Kit, according to manufacturer's instructions (Invitrogen, catalogue no. L7012). An overnight *S. pseudintermedius* ST71 culture was washed with fresh TSB and OD adjusted to 0.5 under treatment with increasing concentrations of extract (0, 0.8, 1.0, 1.6, 3.1, 12.5 µg/ml). After incubation for 4 hr, the bacteria were harvested, washed and resuspended in 1 X PBS. Equal volumes of SYTO9 and propidium iodide (PI) were mixed and 3 µl was added to each sample to a final volume of 1 ml and incubated at room temperature in the dark for 15 min. Flow cytometry measurements were taken on a BioRad ZE5 Cell Analyzer with forward and side scatter parameters for detecting bacteria that were non-stained. Total bacteria were gated by dual stained but SYTO9-positive only population. The percentage dead or membrane-compromised bacteria were detected and recorded by the population that were SYTO9- and PI-positive. Analysis was performed using FlowJo V10 software (BD Biosciences).

## Transmission electron microscopy

*S. pseudintermedius* ST71 cell pellets were immersed in modified Karnovsky's fixative (2 % glutar-aldehyde and 2 % paraformaldehyde in 0.10 M sodium cacodylate buffer, pH 7.4) for at least 4 hr and further postfixed in 1 % osmium tetroxide in 0.1 M cacodylate buffer for 1 hr on ice. The cells were stained all at once with 2 % uranyl acetate for 1 hr on ice, then dehydrated in a graded series of ethanol (50–100%) while remaining on ice. The cells were washed with 100 % ethanol and washed twice with acetone (10 min each) and embedded with Durcupan. Sections were cut at 60 nm on a Leica UCT ultramicrotome and picked up on 300 mesh copper grids. Sections were post-stained with 2 % uranyl acetate for 5 min and Sato's lead stain for 1 min. Grids were viewed using a JEOL JEM-1400Plus (JEOL, Peabody, MA) transmission electron microscope and photographed using a Gatan OneView 4 K digital camera (Gatan, Pleasanton, CA).

## Mouse skin colonization and infection with *S. pseudintermedius*

### mouse skin colonization

All experiments involving live animal work were performed in accordance with the approval of the University of California, San Diego Institutional Animal Care and Use Guidelines (protocol no. S09074). For mouse skin challenge experiments involving *S. felis* strains and *S. pseudintermedius* ST71, the dorsal skin of hairless age-matched 8–10 week-old SKH1 mice (n = 2, per treatment) were scrubbed with alcohol wipes and $5 \times 10^6$/cm² or $5 \times 10^7$/cm² CFU of overnight cultured *S. felis* C4, *S. felis* ATCC 49168 or *S. pseudintermedius* ST71 was inoculated onto 1 × 1 cm sterile gauze pads, which were placed onto the dorsal skin and secured with wound dressing film (Tegaderm [3 M]) for 72 h. For the experiments involving the live *S. felis* C4 bacteria and the extract topical treatment, the dorsal skin of age-matched 8–10 week-old C57BL/6 mice (n = 4, per treatment) were shaved and depilated by using Nair cream followed by removal with alcohol wipes. The skin was allowed to recover from hair removal for at least 24 h before the application of bacteria. Prior to bacterial challenge, the dorsal skin was tape-stripped and *S pseudintermedius* ST71 agar disks (3 % TSB, 2 % agar; diameter 8 mm) containing $5 \times 10^7$ CFU was applied to the skin for 48 h, as previously described (*Nakatsuji et al., 2016*). The dorsal skin and agar disk were covered with Tegaderm and a bandage was applied to hold the agar disk (or gauze later) in place for the duration of the treatment. The bandage, Tegaderm and agar disk were removed and $5 \times 10^7$/cm² CFU of overnight cultured *S. felis* C4 (washed 2X in TSB), or 100 µl of extract (10 mg/ml) or 3 % TSB control, was inoculated onto 2 × 2 cm sterile gauze pads and applied to the infected site every 24 hr for 72 hr. After the treatment of mouse skin with live bacteria or extract, the dressing film and gauze pad were removed, and surface bacteria were collected using a swab soaked in TSB-glycerol solution. The swab head was then placed in 1 mL of TSB-glycerol solution, vortexed (30 s), serial-diluted, and plated onto Baird Parker agar plates supplemented with egg yolk tellurite for enumeration of coagulase-positive staphylococci (*Carter, 1960*) or mannitol salt agar plates for enumeration of all surface staphylococci (*Parisi and Hamory, 1986*).

## Mouse infection model

The day prior to bacterial infection, the dorsal skin of age-matched 8–10 week old C57BL/6 mice (n = 5, per treatment) was shaved and depilated by using Nair cream followed by removal with alcohol wipes. A 50 µl inoculum suspension containing $1 \times 10^7$ CFU of late log phase *S. pseudintermedius* ST71 in PBS was intradermally injected into the dorsal skin using 0.3 mL/31-gauge insulin syringe (BD, Franklin Lakes, NJ). At 1 hr post-infection, a 50 µl suspension of the S. *felis* C4 extract (250 µg, at a concentration of 5 mg/ml in 25 % DMSO) or a control suspension of 1 X PBS in 25 % DMSO was injected twice in two separate skin sites directly adjacent to the bacterial injection site. Body weights of the mice were measured before and after infection every day for 14 days. To determine lesion size, a ruler was positioned adjacent to the mouse skin lesions and digital photos were taken daily with a Kodak PIXPRO Astro Zoom AZ421 and analyzed via ImageJ software (National Institutes of Health Research Services Branch, Bethesda, MD, USA). Lesion size in $mm^2$ was measured by calculating the length x width.

## Quantitative real-time PCR

mRNA transcript abundance was measured by qPCR in NHEKs stimulated with or without TLR2/6 agonist MALP-2 (200 ng/ml) or TLR3 agonist Poly I:C (0.4 µg/ml) in the presence or absence of *S. felis* C4 extract, PSMβ2 or PSMβ3 (10 µg/ml) or DMSO control (0.1%) at 4 hr post-treatment. RNA was extracted from NHEK cells using Pure Link RNA isolation kit (Thermo Fisher) according to manufacturer's instructions. RNA was quantified on a Nanodrop 2000/200c spectrophotometer (Thermo Fisher, USA). Purified RNA ( 1 µg) was used to synthesize cDNA using the iScript cDNA Synthesis Kit (Bio-Rad, USA). Pre-Developed SYBR-Green gene expression assays (Integrated DNA Technologies, USA) were used to evaluate mRNA transcript levels.

## RNA sequencing

NHEK cells were treated with DMSO (0.1%) control, PSMβ2 (10 µg/ml), Poly I:C (0.4 µg/ml) or PSMβ2 and Poly I:C combined, all in triplicate wells, for 4 hr or 24 hr in and RNA was extracted using the PureLink RNA mini kit and triplicate samples were pooled together for each treatment. Isolated RNA was submitted to the UCSD IGM Genomics Center for RNA-sequencing performed on a high-output run V4 platform (Illumina, USA) with a single read 100 cycle runs. Data alignment was performed using Partek Flow genomic analysis software (Partek, USA) with Tophat2 (version 2.0.8) Gene ontology (GO) enrichment analysis was performed on differentially regulated genes ( $\geq$ 1.5 -fold) using DAVID 6.8 or Metascape (*Zhou et al., 2019*).

## Statistical analysis

Significant differences between the means of the different treatments were evaluated using GraphPad Prism version 7.03 (GraphPad Software, Inc, La Jolla, CA). Either unpaired, two-tailed Student's *t* test or one-way analysis of variance (ANOVA) followed by Dunnett's or Tukey's multiple comparisons test were used for statistical analysis and indicated in the respective figure legends. Differences were considered statistically significant with a p value of < 0.05.

# Acknowledgements

The authors acknowledge the participants for their assistance in this project. We thank Ying Jones of the UCSD/CMM electron microscopy facility for TEM sample preparation and Timothy Meerloo for imaging assistance. The EM facility is supported by NIH equipment grant 1S10OD023527. We thank Nina J Gao for providing strains. We thank Jamie Boehmer for her very helpful edits.

# Additional information

**Competing interests**

Kate A Worthing: Dr. Worthing is a co-inventor of technology described in this manuscript that has been disclosed to the University of California San Diego.. Richard L Gallo: is a co-founder, scientific advisor, consultant and has equity in MatriSys Biosciences and is a consultant, receives income and has equity in Sente. The other authors declare that no competing interests exist.

## Funding

| Funder | Grant reference number | Author |
| --- | --- | --- |
| National Institute of Diabetes and Digestive and Kidney Diseases | T32 DK007202 | Robert H Mills |
| National Institutes of Health | R37AI052453 | Richard L Gallo |
| National Institute of Health | R01AR069653 | Richard L Gallo |
| National Institute of Health | R01AR074302 | Richard L Gallo |

The funders had no role in study design, data collection and interpretation, or the decision to submit the work for publication.

## Author contributions

Alan M O'Neill, Conceptualization, Data curation, Formal analysis, Investigation, Methodology, Project administration, Validation, Visualization, Writing - original draft, Writing - review and editing; Kate A Worthing, Conceptualization, Data curation, Formal analysis, Investigation, Methodology, Resources, Validation, Writing - review and editing; Nikhil Kulkarni, Formal analysis, Investigation, Software, Writing - review and editing; Fengwu Li, Investigation, Methodology, Writing - review and editing; Teruaki Nakatsuji, Methodology, Resources, Writing - review and editing; Dominic McGrosso, Gayathri Kalla, Formal analysis, Methodology, Visualization; Robert H Mills, Formal analysis, Investigation, Methodology, Resources, Software, Writing - review and editing; Joyce Y Cheng, Resources, Writing - review and editing; Jacqueline M Norris, Conceptualization, Resources, Supervision, Writing - review and editing; Kit Pogliano, Joe Pogliano, Investigation; David J Gonzalez, Resources, Supervision, Writing - review and editing; Richard L Gallo, Conceptualization, Funding acquisition, Project administration, Resources, Supervision, Writing - review and editing

## Author ORCIDs

Alan M O'Neill (iD) http://orcid.org/0000-0002-5892-6477
Kate A Worthing (iD) http://orcid.org/0000-0001-8713-7189
Kit Pogliano (iD) http://orcid.org/0000-0002-7868-3345
Richard L Gallo (iD) http://orcid.org/0000-0002-1401-7861

## Ethics

All experiments involving live animal work were performed in accordance with the approval of the University of California, San Diego Institutional Animal Care and Use Guidelines (protocol no. S09074).

## Decision letter and Author response

Decision letter https://doi.org/10.7554/eLife.66793.sa1
Author response https://doi.org/10.7554/eLife.66793.sa2

# Additional files

## Supplementary files
• Transparent reporting form

## Data availability

The RNA Sequencing data has been deposited in Dryad with a unique DOI identifier provided: https://doi.org/10.6076/D10019.

The following dataset was generated:

| Author(s) | Year | Dataset title | Dataset URL | Database and Identifier |
|---|---|---|---|---|
| O'Neill AM | 2021 | Antimicrobials from a feline skin commensal bacterium inhibit skin colonization and infection by drug-resistant S. pseudintermedius | https://doi.org/10.6076/D10019 | Dryad Digital Repository, 10.5061/dryad/D10019 |

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
