## [Decision Letter]

**Acceptance summary:**

You provide here characterization of how a skin commensal from cats produces phenol soluble modulin β (PSM-B) antimicrobial peptides with activities against gram positive organisms and particularly Staphylococcus in vitro and in an in vivo murine model of skin infection. The antimicrobial agents appear to target the cell envelope of the pathogen while also showing anti-inflammatory effect on keratocytes. Thus, your work constitutes a proof of principle of a new potential topical therapeutic option against skin Staphylococcus infections.

**Decision letter after peer review:**

Thank you for submitting your article "Antimicrobials from a feline commensal bacterium inhibit skin infection by drug-resistant *S. pseudintermedius*" for consideration by *eLife*. Your article has been reviewed by 3 peer reviewers, and the evaluation has been overseen by a Reviewing Editor and Y M Dennis Lo as the Senior Editor. The following individual involved in review of your submission has agreed to reveal their identity: Andrew M Edwards (Reviewer #1).

Essential revisions:

A few points need addressing before the study can be considered for publication:

– It is not clear how much PSM-B is in the extract used in the mouse studies. The MIC of the extract was 8 ug/ml but purified PSM-B did not significantly inhibit bacterial growth until 25-50 ug/ml. This disparity suggests that something else in the extract might be responsible for the antibacterial activity of the extract. There is a possibility that the active molecule(s) have not been fully identified.

(1) The MIC for each peptide is provided (50 ug/ml), but since these are produced as a mixture by *S. felis* please test whether they show synergistic activity. This would explain why the MIC of the native supernatant extract was much lower.

(2) Use PSM-B alone in in vivo experiments to convincingly test if it is the active fraction of the C4 extract or delete the corresponding gene and show loss of competitive activity or overexpress it from a non-competitive strain and show gain of function.

– The characterization of the activity of the extract as an anti-membrane molecule needs strengthening. Please provide new data for Figure 3 B-D with a time course (t0 to t1hour) and more controls including cell wall, translation, and membrane targeting antibiotics such as daptomycin and CCCP as a chemical inducing membrane permeability and collapse of ATP levels. Show percentages of dead stain + cells as measured by Flow Cytometry.

– In order to strengthen the conclusion that PSM-B have no activity on NHEKs, provide cytotoxicity measures for higher concentrations than 100ug/ml which is the first concentration at which bacterial growth is totally blocked.*Reviewer #1 (Recommendations for the authors):*

The authors set out to identify bacteria that could outcompete *Staphylococcus pseudointermedius*, a leading cause of skin infection and exacerbation of atopic dermatitis in companion animals and increasingly in humans. Antibiotic resistance, the off-target effects of antibiotic use and poor efficacy of topical approaches necessitates the development of better therapeutics for combatting this pathogen.

The authors used appropriate methods to identify and characterise competitive bacteria, followed by determination of the underlying mechanism of competition. The authors then show that a strain of *S. felis* could succesfully be used to treat skin infection caused by *S. pseudointermedius*. The aims were achieved and the conclusions are justified by the results.

Strengths: The approaches used, with key findings typically supported by more than one method. For example, the use of synthetic peptides to confirm data from spent culture supernatant. The findings are very compelling, with good efficacy shown in relevant animal models.

Weakness: activity is shown against a single *S. pseudointermedius* strain, albeit one that is representative of the predominant lineage.

The use of bacteria as therapeutics is developing rapidly as a field and this work makes a very useful contribution. If this can be shown to be useful in animals it will provide a strong platform for larger development for human use. The approach described is fairly standard, but useful to know that it worked.*Reviewer #2 (Recommendations for the authors):*

The manuscript by O'Neill et al., describes how a skin commensal isolated from cats, *S. felis* produces an antimicrobial that is active against *S. aureus*. Importantly, use of *S. felis* or an extract containing the antimicrobial was capable of improving the outcome of a mouse skin and soft tissue infection. The researchers identified PSM-B as the molecule predominantly responsible for the anti-Staphylococcal activity and interestingly describe an additional anti-inflammatory activity associated with the molecule. Novel antimicrobials against *S. aureus* are desperately needed and the identification of bacterial species that can outcompete or inhibit *S. aureus* as well as the identification of specific molecules with low toxicity and anti-Staphylococcal activity is of major importance. The paper is well written, clear and concise. The experiments appear to have been carefully carried out.

I have some concerns as follows:

1. It is not clear how much PSM-B is in the extract used in the mouse studies. The data presented do not convincingly show that the in vivo activity is truly orchestrated by PSM-Bs. The MIC of the extract was 8 ug/ml but purified PSM-B did not completely inhibit bacterial growth until 100 ug/ml. This disparity suggests that something else in the extract was responsible for the antibacterial activity or is an essential component. With this in mind, there is a possibility of a critical gap in this manuscript, where the most impressive activity is shown in figure 2, using extracts and bacterial producers and then a leap is made to PSM-Bs as the driver of this activity. I do not think the data supports this, meaning the active molecule(s) have not been identified, which significantly reduces the impact of this work.

2. The membrane activity is not fully supported. The assays measuring ROS and loss of ATP could be easily associated with a plethora of antibiotics.

3. Although the authors describe the antimicrobial activity of PSM-B and the lack of toxicity, I worry the authors are over-interpreting their data here. In figure 4 A and B, you only see full inhibition of bacterial growth at the same concentration where cytotoxicity becomes apparent. The authors did not examine any higher concentration. As these assays are very different, with one measuring ability of bacteria to proliferate and the other measuring release of lactate dehydrogenase from eukaryotic cells, it's possible that PSM-Bs have similar activity against both bacterial and eukaryotic membranes, which would severely limit their utility.*Reviewer #3 (Recommendations for the authors):*

The authors assessed the effect of 85 staphylococcal carriage isolates from dogs and cats on a currently successful, multidrug-resistant canine skin and soft-tissue pathogen, MRSP ST71 through a large number of in vitro and in vivo (mouse skin) assays. One particular Staphylococcus felis isolate (C4) was identified in co-culture/agar inhibition experiments. Subsequently, the ability of this isolate (or its extract containing antimicrobial peptides) to inhibit MRSP was tested on mice skin. Electronmicroscopy was used to demonstrate a cell membrane-disrupting effect. Antimicrobial peptides were purified from the extract and tested again for their inhibitory effect. Lastly, immune-responses were tested in various assays, including human keratinocytes.

The current presentation of the manuscript makes it highly challenging to read at the moment but study contains a substantial amount of findings that might eventually lead to further development of new alternatives for antimicrobial treatment of skin infections.

---

## [Author Response]

Essential revisions:A few points need addressing before the study can be considered for publication:– It is not clear how much PSM-B is in the extract used in the mouse studies. The MIC of the extract was 8 ug/ml but purified PSM-B did not significantly inhibit bacterial growth until 25-50 ug/ml. This disparity suggests that something else in the extract might be responsible for the antibacterial activity of the extract. There is a possibility that the active molecule(s) have not been fully identified.

We thank the reviewers for highlighting this discrepancy and we have conducted a series of additional experiments to explain why the extract has higher potency. A genetic analysis of *S. felis* C4 identified a biosynthetic gene cluster with similarity to an antimicrobial thiopeptide encoding micrococcin P1. The presence of micrococcin P1 in the *S. felis* extract was confirmed by HPLC and mass spectrometry (Figure 4 and Figure 4—figure supplement 1). We believe the presence of micrococcin P1 explains the discrepancy in killing as a synthetic micrococcin π exhibited a similar MIC range compared to the *S. felis* extract and was shown to precipitate into butanol. We have also performed additional MOA studies on the *S. felis* C4 extract and demonstrated a bacterial cytological profile with striking similarities to cells treated with micrococcin P1 (Figure 5). As a result of this extensive new data, we now provide a substantially revised manuscript that describes this additional, potent antimicrobial produced by *S. felis* C4.

(1) The MIC for each peptide is provided (50 ug/ml), but since these are produced as a mixture by S. felis please test whether they show synergistic activity. This would explain why the MIC of the native supernatant extract was much lower.

We thank the reviewers for this excellent suggestion. We conducted the experiment and found no evidence of a synergistic activity of the PSMs in combinations versus the same concentration of PSMs alone. We provide this new data as an additional supplementary figure (Figure 3—figure supplement 3) in the manuscript. In addition, during our initial investigations into why the PSMs were not providing MICs similar to the extract, we speculated that the N-terminal formylation of the starting methionine may negatively impact activity in vitro. Studies of PSMs produced by *S. aureus* have revealed the presence of native peptides with or without the N-formylation. For that reason we also generated a non-formylated synthetic PSMB2 and tested its bioactivity. Overall, there was no significant difference in the presence or absence of N-formylation in the inhibition of MRSP growth or in terms of synergistic activity (Figure 3—figure supplement 3). We have now added that data point back into the original Figure 4A, which is now presented as Figure 3H in our current revised version.

2) Use PSM-B alone in in vivo experiments to convincingly test if it is the active fraction of the C4 extract or delete the corresponding gene and show loss of competitive activity or overexpress it from a non-competitive strain and show gain of function.

We thank the reviewers for this excellent suggestion. We made extensive efforts to genetically manipulate *S. felis* C4 but found it to be unamenable to transformation with the current tools used for staphylococci mutagenesis. We used empty vectors pJB38 and pKOR1 that had worked well in our lab previously, generating successful transformants in strains belonging to S. epidermidis and S. hominus. Some of the conditions we optimized include:

1. Preparation of highly pure and concentrated plasmid preps in DC10b *E. coli*

2. Fresh versus frozen competent *S. felis* C4 cells

3. Electroporation with different concentrations of *S. felis* C4 at early or mid exponential growth

4. Electroporation of *S. felis* C4 with different amounts of plasmid DNA (.1 – 10 ug)

5. 1900kV, 2100kV and 2300kV electroporation with different amounts of plasmid DNA in 1mm or 2mm cuvettes

6. 45-55 deg heat shock followed by electroporation to inactivate host restriction modification system in *S. felis* C4 as performed by Lofblom et al., 2006. J. Applied Microbiology.

7. Post electroporation recovery in 500mM sucrose for different time periods (1h-3h). Lastly, we show a representative image of our transformation experiments. Whilst we were able to successfully generate plasmid transformants with the positive control S. epidermidis 1457 strain (left image of ). We were never able to successfully recover transformants for the *S. felis* C4 strain.

Due to the failure to mutate the PSM-encoding genes in *S. felis* C4, we investigated if we could recapitulate our in vivo data that showed the *S. felis* C4 extract improved MRSP infection using purified PSM peptides. We followed the same process as before and injected 500 μg of PSM adjacent to the infected skin lesion and measured lesion size over time. Unfortunately, the PSMs did not result in a significant reduction in lesion size. This is not surprising given the higher MIC values for PSM versus the *S. felis* extract. However, in light of our new data identifying the presence of micrococcin P1 in the extract, we speculate a beneficial effect or applying micrococcin to infected skin wounds. It has been reported in the literature that micrococcin P1 can reduce MRSA bacterial burden in infected mouse skin (Liu et al., 2020. Microbiome).

– The characterization of the activity of the extract as an anti-membrane molecule needs strengthening. Please provide new data for Figure 3 B-D with a time course (t0 to t1hour) and more controls including cell wall, translation, and membrane targeting antibiotics such as daptomycin and CCCP as a chemical inducing membrane permeability and collapse of ATP levels. Show percentages of dead stain + cells as measured by Flow Cytometry.

We have now provided detailed mechanism of action studies for the *S. felis* C4 by employing a powerful approach called bacterial cytological profiling (BCP). This approach generates a cytological profile of bacteria treated with different classes of antibiotics that target distinct cellular pathways. Our new data (Figure 5A) shows that cells treated with *S. felis* C4 extract or tetracycline reveal the presence of condensed nucleoids with toroidal structures, indicating a target of protein translation. Moreover, our panel EM images (now Figure 5B) had previously highlighted evidence of condensed chromosomes, in addition to the substantial alteration in the cell membrane, validating the findings from BCP. As suggested by the reviewers we performed time course experiments to measure the levels of ATP with additional antibiotic controls. However, in light of our more powerful BCP approach, which already includes antibiotic controls with different MoA, we opted to leave the ATP and ROS dose-response experiment in the main figure. Nevertheless, we now include results from the LIVE/DEAD assay providing the percentages of PI-positive cells from flow cytometry and removed the previous representative microscopy images.

– In order to strengthen the conclusion that PSM-B have no activity on NHEKs, provide cytotoxicity measures for higher concentrations than 100ug/ml which is the first concentration at which bacterial growth is totally blocked.

This data is now provided in Figure 2H.